# Surmising synchrony of sound and sight: Factors explaining variance of audiovisual integration in hurdling, tap dancing and drumming

**Nina Heins**[1,2☯], **Jennifer Pomp**[1,2☯], **Daniel S. Kluger**[2,3], **Stefan Vinbrüx**[4], **Ima Trempler**[1,2], **Axel Kohler**[2], **Katja Kornysheva**[5], **Karen Zentgraf**[6], **Markus Raab**[7,8], **Ricarda I. Schubotz**[1,2]*

1 Department of Psychology, University of Muenster, Muenster, Germany, 2 Otto Creutzfeldt Center for Cognitive and Behavioral Neuroscience, University of Muenster, Muenster, Germany, 3 Institute for Biomagnetism and Biosignal Analysis, University Hospital Muenster, Muenster, Germany, 4 Institute of Sport and Exercise Sciences, Human Performance and Training, University of Muenster, Muenster, Germany, 5 School of Psychology and Bangor Neuroimaging Unit, Bangor University, Wales, United Kingdom, 6 Department of Movement Science and Training in Sports, Institute of Sport Sciences, Goethe University Frankfurt, Frankfurt, Germany, 7 Institute of Psychology, German Sport University Cologne, Cologne, Germany, 8 School of Applied Sciences, London South Bank University, London, United Kingdom

☯ These authors contributed equally to this work.
* rschubotz@uni-muenster.de

**Data Availability Statement:** All files are available from the OSF database: Schubotz, Ricarda. 2020. "AVIA - Audiovisual Integration in Hurdling, Tap

## Abstract

Auditory and visual percepts are integrated even when they are not perfectly temporally aligned with each other, especially when the visual signal precedes the auditory signal. This window of temporal integration for asynchronous audiovisual stimuli is relatively well examined in the case of speech, while other natural action-induced sounds have been widely neglected. Here, we studied the detection of audiovisual asynchrony in three different whole-body actions with natural action-induced sounds–hurdling, tap dancing and drumming. In Study 1, we examined whether audiovisual asynchrony detection, assessed by a simultaneity judgment task, differs as a function of sound production intentionality. Based on previous findings, we expected that auditory and visual signals should be integrated over a wider temporal window for actions creating sounds intentionally (tap dancing), compared to actions creating sounds incidentally (hurdling). While percentages of perceived synchrony differed in the expected way, we identified two further factors, namely high event density and low rhythmicity, to induce higher synchrony ratings as well. Therefore, we systematically varied event density and rhythmicity in Study 2, this time using drumming stimuli to exert full control over these variables, and the same simultaneity judgment tasks. Results suggest that high event density leads to a bias to integrate rather than segregate auditory and visual signals, even at relatively large asynchronies. Rhythmicity had a similar, albeit weaker effect, when event density was low. Our findings demonstrate that shorter asynchronies and visual-first asynchronies lead to higher synchrony ratings of whole-body action, pointing to clear parallels with audiovisual integration in speech perception. Overconfidence in the naturally expected, that is, synchrony of sound and sight, was stronger for intentional

Dancing and Drumming." OSF. October 2. osf.io/
ksma6. " The DOI is 10.17605/OSF.IO/KSMA6.

**Funding:** Author: RIS; Grant number: SCHU1439/
4-2; Name of funder: German Research Foundation
(DFG, Deutsche Forschungsgemeinschaft); URL:
https://www.dfg.de/en/index.jsp. The funders had
no role in study design, data collection and
analysis, decision to publish, or preparation of the
manuscript.

**Competing interests:** The authors have declared
that no competing interests exist.

(vs. incidental) sound production and for movements with high (vs. low) rhythmicity, presumably because both encourage predictive processes. In contrast, high event density appears to increase synchronicity judgments simply because it makes the detection of audiovisual asynchrony more difficult. More studies using real-life audiovisual stimuli with varying event densities and rhythmicities are needed to fully uncover the general mechanisms of audiovisual integration.

## Introduction

From simple percepts like the ticking of a clock to complex stimuli like a song played on a guitar–in our physical world we usually perceive visual and auditory components alongside each other. The multisensory nature of our world has many advantages–it increases the reliability of sensory signals [1] and helps us navigate noisy environments, e.g. when one of the senses is compromised [2]. On the other hand, multimodality poses a challenge to our brains. Percepts from different senses have to be monitored to decide whether they belong to the same event and have to be integrated or segregated.

The impression of unity, i.e. the feeling that percepts belong to the same event, depends on many factors [3], one of them being the temporal coincidence of stimuli. For instance, we usually perceive visual and auditory speech as occurring at the same time, although these signals differ both in their neural processing time (10 ms for auditory signals vs. 50 ms for visual signals) and their physical "travel time" (330 m/sec for auditory signals, 300.000.000 m/sec for visual signals). Indeed, there seems to be a temporal window for the integration of audiovisual signals (temporal binding window: e.g. [2, 4]. Although the much cited notion that visual speech naturally leads auditory speech [5] has been recently revised [6], the temporal binding window seems to favor the visual channel leading the auditory channel. This is reflected in audio-first asynchronies (where the auditory signal leads the visual signal) being detected at smaller delays than visual-first asynchronies (where the visual signal leads the auditory signal: e.g. [7]. Also, the so-called McGurk effect—an illusion where the perception of an visual speech component and a different auditory speech component leads to the perception of a third auditory component [8]—is prevalent for larger visual-first than audio-first asynchronies [2]. This effect is suggested to show that visual speech acts as a predictor for auditory speech [9]. Visual speech aids auditory speech recognition even when visual and auditory signals are asynchronous, up to the boundaries of the temporal binding window [10]. Consequently, a coherent perception can be maintained for relatively large temporal asynchronies [7].

Although generally asymmetric, the width of temporal binding windows depends on different stimulus properties. For instance, this width seems to be up to five times wider for speech signals compared to simple flash and beep stimuli [11], more symmetrical for speech [12] and generally wider for more complex stimuli [4]. Experience seems to shape the width of the temporal binding window as well: Musicians have narrower temporal binding windows [13] and the window can be widened when participants are continuously presented with asynchronous stimuli [11, 14].

Notably, research on the audiovisual perception has so far focused on speech [2, 9, 15, 16], whereas other types of stimuli have been largely neglected [17]. There are only a few studies looking at the audiovisual perception of musical stimuli [18–20] and object-directed actions, e.g. a hammer hitting a peg [21], a soda can being crashed [22], hand claps and spoon taps [23], and a chess playing sequence [24]. These studies mostly find that non-speech sounds

have a narrower temporal binding window than speech, i.e. asynchronies are detected at smaller temporal delays. This is explained by the more predictable moments of impact [24] which is also in line with a better asynchrony detection for the more visually salient bilabial speech syllables [25].

Although audiovisual integration of our own and other people's actions is omnipresent in our everyday life the same way speech is, it is not nearly as well explored. It is an open issue whether effects that have been observed for the audiovisual integration in language and music generalize to the breadth of self-generated sounds we are familiar with [17]. Also, aberrant audiovisual integration in psychiatric diseases [26] and neurological impairments [27] may well apply beyond speech and music, and thus affect the perception and control of own action. To fully understand audiovisual integration, we need to consider this phenomenon in its entire range, from human-specific speech and music to sounds that we, as all animals, generate just by moving and contacting the environment.

The two studies we present here were motivated by the observation that speech and music are both actions that generate sounds *intentionally*. Moreover, both speech and musical sounds score particularly high on two further properties: *event density* and *rhythmicity*. Therefore, in order to examine the potential generalizability of audiovisual integration from these domains to other natural sound-inducing actions, we were interested to find out whether incidentally action-induced sounds would show comparable patterns of audiovisual integration as intentionally action-induced sounds (Study 1); and whether audiovisual integration is modulated by variable event density and rhythmicity (Study 2). In two recent fMRI studies, we observed that brain networks for processing intentionally produced sounds differ from those for incidentally produced action sounds. Interestingly, rather than triggering higher auditory attention, intentional sound production more so than incidental sound production encouraged predictive processes leading to the typical attenuation pattern in primary auditory cortex [28, 29].

## Study 1

In the first study, we used two types of non-speech auditory stimuli created by whole-body actions, namely hurdling and tap dancing. We decided to use two different types of sporting action that allowed us to study the processing of natural movement sounds in an ecologically valid context. This also had the particular advantage that the subjects' attention was not directed in any direction, since we created a completely natural perceptual situation. We applied a total of eight different asynchronies (ranging from a 400 ms lead of the auditory track to a 400 ms lead of the visual track) and a synchronous condition with a simultaneity judgment task. In addition to using new, more ecologically valid stimuli, we examined the influence of intentionality of sound generation. We have the intention to generate sounds by a tap dancing action (just like speaking, or playing a musical instrument, etc.), while sounds generated by a hurdling action (or by placing a chess piece on the board, for instance) are rather an incidental by-product of the action. Based on a previous study [28], demonstrating that the cerebral and behavioral processing of action-induced sounds significantly differs for intentionally and incidentally generated sounds, perceived audiovisual synchrony of tap dancing stimuli may yield similar effects as speech stimuli, and hurdling similar effects as object-directed actions.

Accordingly, we set out to test the following specific hypotheses: We expected shorter asynchronies to be generally perceived as synchronous more often than longer asynchronies (Hypothesis 1). Additionally, we expected visual-first asynchronies to be perceived as synchronous more often than corresponding audio-first asynchronies in both types of action (Hypothesis 2). Moreover, suggesting that tap dancing is comparable to speech production in having a

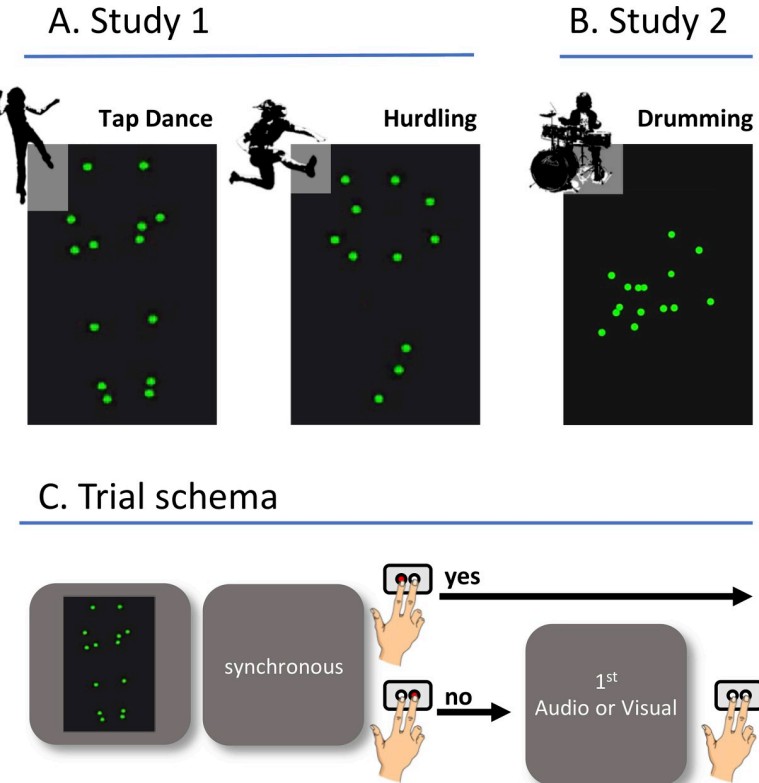

**Fig 1. Stimuli and task.** Screenshots of the stimuli used in (A.) Study 1 and (B.) Study 2. The lower panel (C.) shows a schema of the trial and required responses. Participants were presented with videos showing PLDs of hurdling, tap dancing, or drumming. Subsequently, they were asked to judge, in a dual forced choice setting, whether the audiovisual presentation was synchronous or not. In case of a negative decision, participants had to furthermore judge whether sound was leading video or vice versa.

larger temporal binding window than incidentally produced action sounds, we expected that this synchrony bias vanishes for the longer delays in hurdling but still persists for tap dancing (Hypothesis 3). This should manifest in significant differences between visual-first and audio-first delays in the larger delay types (i.e. 320–400 ms) in tap dancing, but not in hurdling.

## Materials and methods–Study 1

### Participants

The sample consisted of 22 participants (12 males, 10 females) with an age range from 20 to 32 years ($M$ = 23.9, $SD$ = 2.9), including only right-handers. We recruited only participants who never had a training in tap dancing or hurdling. Participants signed an informed consent explaining the procedure of the experiment and the anonymity of the collected data. Participants studying psychology received course credit for their participation. The study was approved by the Local Ethics Committee at the University of Münster, Germany, in accordance with the Declaration of Helsinki.

### Stimuli

The stimuli used in this study stem from a previous fMRI study [28] and consisted of point-light displays (PLDs) of hurdling and tap dancing with their matching sounds (Fig 1A; see also

Supplementary Material for exemplary videos). Note that tap dancing and hurdling share a basic property, that is, all sounds generated by these actions are caused by foot-ground contact. Fourteen passive (retroreflective) markers placed symmetrical on the left and the right shoulders, elbows, wrists, hip bones, knees, ankles, and toes (over the second metatarsal head). Nine optical motion capture cameras (Qualisys opus 400 series) of the Qualisys Motion Capture System (https://www.qualisys.com; Qualisys, Gothenburg, Sweden) were used for kinematic measurements. The sound generated by hurdling was recorded using in-ear microphones (Soundman OKM Classic II) and by a sound recording app on a mobile phone for tap dancing. The mobile phone was hand-held by a student assistant sitting about one meter behind the tap dancing participant.

After recording, PLDs were processed using the Qualisys Track Manager software (QTM 2.14), ensuring visibility of all 14 recorded point-light markers during the entire recording time. Sound data were processed using Reaper v5.28 (Cockos Inc., New York, United States). In a first step, stimulus intensities of hurdling and tap dancing recordings were normalized separately. In order to equalize the spectral distributions of both types of recordings, the frequency profiles of hurdling and tap dancing sounds were then captured using the Reaper plugin Ozone 5 (iZotope Inc, Cambridge, United States). Finally, the difference curve (hurdling–tap dancing) was used by the plugin's match function to adjust the tap dancing spectrum to the hurdling reference. PLDs and sound were synchronized, and the subsequent videos were cut using Adobe Premiere Pro CC (Adobe Systems Software, Dublin, Ireland). All videos had a final duration of 5.12 seconds. Note that we employed the 0 ms lag condition as an experimental anchor point, being aware that if the observer watched actions from the distance of the camera there would have been a very slight positive lag of audio of about 14 ms. This time lag was the same for both the hurdling and tap dancing stimuli, so that no experimental confound was induced. The final videos had a size of 640x400 pixels, a sampling rate of 25 frames per second and an audio sampling rate of 44 100 Hz. Due to the initial distance between the hurdling participant and the camera system, the hurdling sounds were audible before corresponding PLDs were fully visible. To offset this marked difference between hurdling and tap dancing stimuli in the visual domain, we employed a visual fade-in and fade-out of 1000 ms (25 frames) using Adobe Premiere, while the auditory track was presented without fading.

The stimulus set used here consisted of four hurdling and four tap dancing videos, each of which was presented at nine different "asynchronies" of the sound respective to the PLD (± 400 / 320 / 200 / 120 ms, and 0 ms), with negative values indicating that the audio track was leading the visual track (audio-first) and positive values indicating that the visual track was leading the audio track (visual-first)), resulting in a total of 72 different stimuli (exemplary videos are provided in the Supplementary Material). Asynchrony sizes were chosen based on similar values used in previous studies (e.g. [22, 24]). Finally, prepared videos had an average length of 6 s.

A separate set of 40 hurdling and 40 tap dancing videos with a lag of 0 ms (synchronous) was used to familiarize participants with the synchronous PLDs. All stimuli had a duration of 4000 ms. Videos showed three hurdling transitions for the hurdling stimuli and a short tap dancing sequence for the tap dancing stimuli.

**Acoustic feature extraction: Event density and rhythmicity.** Core acoustic features of the 16 newly recorded drumming videos as well as the 8 original videos from Study 1 were extracted using the MIRtoolbox (version 1.7.2) for Matlab [30]. The toolbox first computes a detection curve (amplitude over time) from the audio track of each video. Form this detection curve, a peak detection algorithm then determines the occurrence of distinct acoustical events (such as the sound of a single step). The number of distinct events per second quantifies the *event density* of a particular recording.

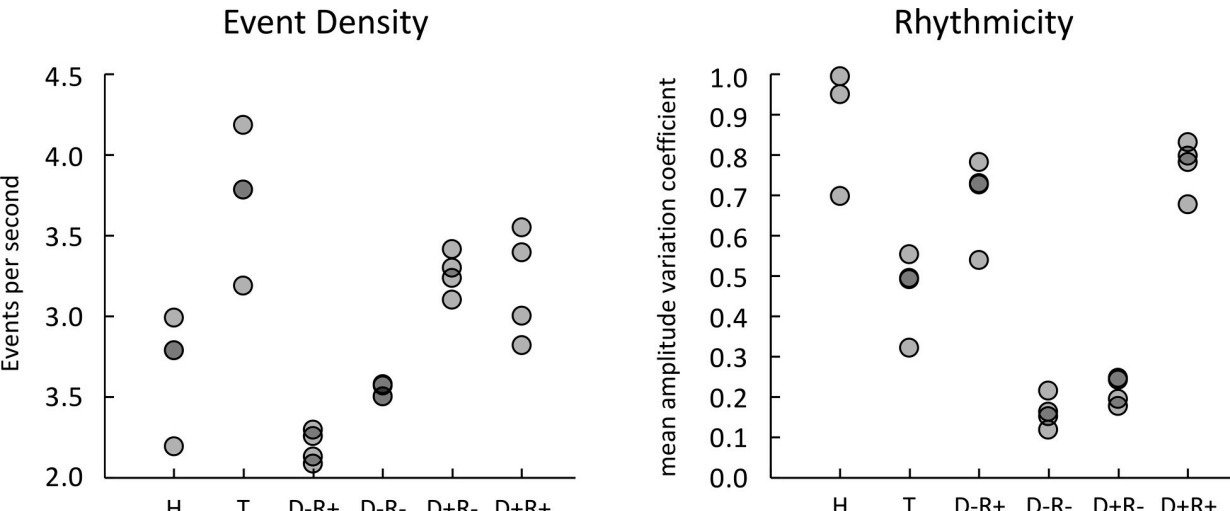

**Fig 2. Auditory stimulus features, Study 1 and 2.** Left panel shows the event density measured in the videos showing hurdling (H), tap dancing (H) (Study 1) and in the four sub-conditions of the drumming videos implementing combinations of high and low event density (D-, D+) and high and low rhythmicity (R+, R-) (Study 2). Each dot represents one recording. Right panel shows a measure of rhythmicity for the same set of recordings, operationalized as the variability of each recording's amplitude envelope. Amplitude variation is shown as the coefficient of variation, i.e. the standard deviation of amplitude normalized by mean amplitude.

Acoustic events vary in amplitude, with accentuated events being louder than less accentuated ones. Therefore, we computed within-recording variance of the detection curve (normalized by the total number of events) to quantify to what extent each recording contained both accentuated and less accentuated events (see Fig 2): A recording with equally spaced, clearly accentuated events was defined as more rhythmic than a recording whose events are more or less equal in loudness (i.e., with low variation between events). An illustrative example of this approach is shown in S1 Fig. To allow comparison of rhythmicity across videos (independently of mean loudness), amplitude variability was computed as the *coefficient of amplitude variation*, i.e. the standard deviation of amplitude divided by its mean.

**Assessment of motion energy (ME).** The overall *motion energy* for hurdling and tap dancing videos was quantified using Matlab (Version R2019b). For each video, the total amount of motion was quantified using frame-to-frame difference images for all consecutive frames of each video. Difference images were binarized, classifying pixels with more than 10 units luminance change as moving and those pixels below 10 units luminance as not moving. Above-threshold ("moving") pixels were finally summed up for each video, providing its motion energy [31]. This approach yielded comparable levels for our experimental conditions, with a mean motion energy of 1189 for hurdling and 1220 for tap dancing (S2 Fig).

## Procedure

The experiment was conducted in a noise-shielded and light-dimmed laboratory. Participants received a short instruction about the procedure of the experiment and signed the informed written consent before the experiment started. Participants were seated with a distance of approximately 75 cm to the computer screen. All stimuli were presented using the Presentation software (Neurobehavioral Systems Inc., CA). Headphones were used for the presentation of the auditory stimuli.

The experiment consisted of four blocks. The first block contained synchronous videos (0 ms lag) to familiarize participants with the PLD. To ensure their attention, participants were

engaged in a cover task during this first block: They were asked to rate, by a dual forced-choice button press (male/female), the assumed gender of the person performing the hurdling or tap dancing action. There were no hypotheses concerning the gender judgment task and this part of the study was not analyzed any further.

Three blocks with the experimental task were presented thereafter. Within each of these blocks, all the 72 stimuli (four hurdling and four tap dancing videos, each with nine different audiovisual asynchronies) were presented twice, resulting in 144 trials per block and 432 trials in total. A pseudo-randomization guaranteed that no more than three videos of the same delay type (audio-first vs. visual-first) were presented in a row to prevent adaptation to one or the other. Additionally, it was controlled that no more than two videos of the same asynchrony were presented directly after each other.

A trial schema of the experimental task is given in Fig 1C. After presentation of each video (4000 ms) participants had to indicate whether they perceived the visual and auditory input as "synchronous" or "not synchronous", pressing either the left key (for synchronous) or the right key (for not synchronous) on the response panel with their left and right index finger. If they decided that picture and sound were "not synchronous", there was a follow-up question concerning the assumed order of the asynchrony ("sound first" or "picture first", corresponding to the delay types *audio-first* and *visual-first*, respectively). We opted for a simultaneity judgment tasks rather than a temporal order judgment, because simultaneity judgment tasks are easier to perform for participants and have a higher ecological validity [32]. Responses were self-paced, but participants were instructed to decide intuitively and as fast as possible. A 1000 ms fixation cross was presented at the middle of the screen before the next video started.

### Experimental design

The study employed a three-factorial within-subjects design. The dependent variable was the percentage of trials perceived as synchronous. Trials with a reaction time above 3000 ms were discarded from the analyses. The first factor was ACTION with the factor levels *hurdling* and *tap dancing*. The different delays were generated by combinations of the factor ASYNCHRONY SIZE (*120 ms*, *200 ms*, *320 ms*, *400 ms*) and ASYNCHRONY TYPE (*audio-first*, *visual first*). Note that all delays where the auditory track was leading the visual track were labeled *audio-first*, while all delays where the visual track was leading the auditory track were labeled *visual-first*. For this analysis, we did not include the 0 ms lag (synchronous) condition, as it could not be assigned to either the *audio-first* or the *visual-first* condition. A 2 x 4 x 2 ANOVA was calculated.

### Results—Study 1

Trials with response times that exceeded 3000 ms were excluded from the analyses (470 out of 9504). Mauchly's test indicated that the assumption of sphericity was violated for ASYNCHRONY SIZE ($x^2(5) = 14.89$, $p = .011$). Therefore, degrees of freedom were corrected using Greenhouse-Geisser estimates of sphericity ($\varepsilon = .72$). Behavioral results are depicted in Figs 3 and 4.

The ANOVA revealed a main effect of ASYNCHRONY SIZE ($F(2.2, 45.3) = 197.96$, $p < .001$). As expected (Hypothesis 1), trials with the 120 ms asynchrony were rated as synchronous significantly more often (M = 68.8%, SD = 11.9%) than trials with the 200 ms asynchrony (M = 53.4%, SD = 14.0%, $t(21) = 8.8$, $p < .001$), which were in turn rated as synchronous more often than trials with the 320 ms asynchrony (M = 34.0%, SD = 11.8%, $t(21) = 13.2$, $p < .001$), and those were rated as synchronous more often than trials with the 400 ms asynchrony (M = 29.5%, SD = 8.7%, $t(21) = 3.7$, $p = .001$).

The main effect of ASYNCHRONY TYPE was significant as well ($F(1, 21) = 198.87$, $p < .001$), with visual-first asynchronies (M = 59.2%, SD = 11.7%) being rated as synchronous significantly

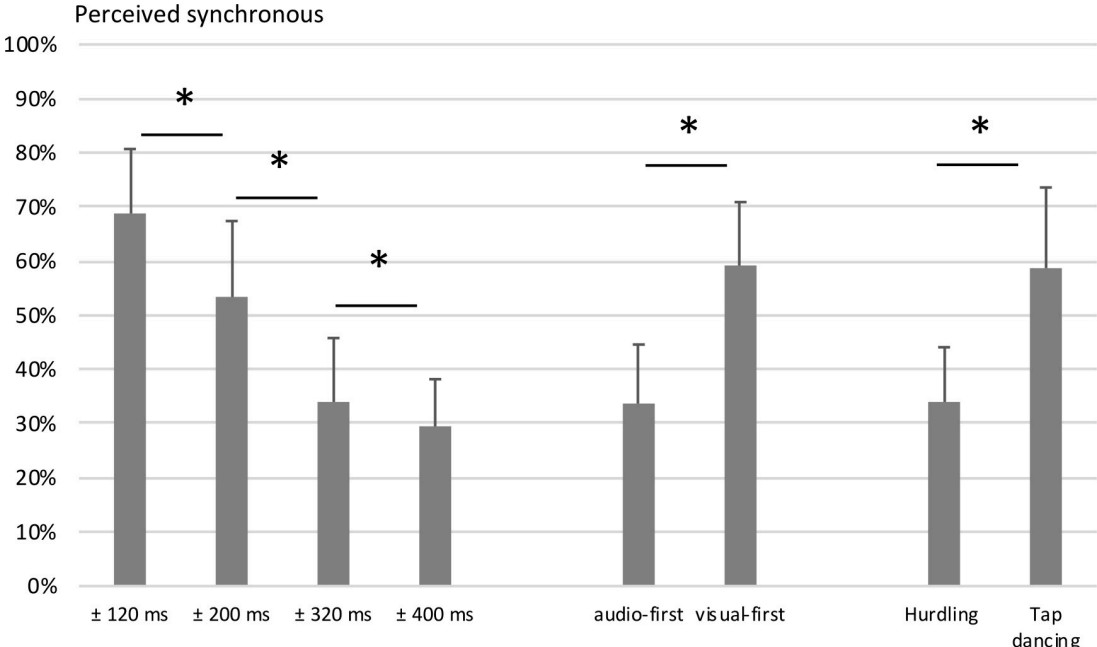

**Fig 3. Main effects of audiovisual (a)synchrony ratings, Study 1.** Displayed are the mean percentages of trials perceived as synchronous, aggregated for the factors asynchrony size, asynchrony type, and action type. Error bars show standard deviations. Statistically significant differences ($p < .001$) are marked with asterisks.

more often than audio-first asynchronies (M = 33.7%, SD = 10.9%), as expected (Hypothesis 2).

Unexpectedly, the main effect of ACTION TYPE was also significant with $F(1, 21) = 64.55$, $p < .001$, driven by overall more synchronous ratings in the tap dancing condition (M = 58.9%, SD = 14.9%) compared to the hurdling condition (M = 34.0%, SD = 10.1%). Note that this finding motivated Study 2, as outlined below.

In line with Hypothesis 3, the interaction of ASYNCHRONY SIZE, ASYNCHRONY TYPE, and ACTION TYPE was significant ($F(3,63) = 10.51$, $p < .001$). Bonferroni-corrected pairwise post-hoc $t$-tests comparing the respective audio-first and visual-first conditions revealed that visual-first conditions in tap dancing were perceived as synchronous more often for the 120 ms asynchrony (M = 88.0%, SD = 11.30%, M = 49.9%, SD = 19.0%, $t(21) = 11.8$, $p < .001$), the 200 ms asynchrony (M = 75.5%, SD = 22.5%, M = 49.4%, SD = 19.8%, $t(21) = 6.0$, $p < .001$), the 320 ms asynchrony (M = 59.5%, SD = 19.8%, M = 44.5%, SD = 18.9%, $t(21) = 5.2$, $p < .001$) and the 400 ms asynchrony (M = 60.1%, SD = 16.2%, M = 44.0%, SD = 16.0%, $t(21) = 4.4$, $p = .001$). In hurdling, visual-first conditions were perceived as synchronous more often than their respective audio-first conditions for the 120 ms asynchrony (M = 91.4%, SD = 10.5%, M = 45.9%, SD = 23.5%, $t(21) = 10.4$, $p < .001$), the 200 ms asynchrony (M = 68.2%, SD = 18.0%, M = 20.6%, SD = 17.1%, $t(21) = 12.6$, $p < .001$), the 320 ms asynchrony (M = 23.1%, SD = 16.7%, M = 9.1%, SD = 8.7%, $t(21) = 4.0$, $p < .001$), but not for the 400 ms asynchrony (M = 7.7%, SD = 9.7%, M = 6.4%, SD = 8.9%, $t(21) = 0.6$, $p = .588$). This was in accordance with our assumption that the visual-first bias is observed even at very long asynchronies for tap dancing but vanishes for hurdling.

Furthermore, the interaction of ACTION TYPE and ASYNCHRONY SIZE ($F(3,63) = 88.71$, $p < .001$) and the interaction of ASYNCHRONY SIZE and ASYNCHRONY TYPE ($F(3,63) = 51.31$, $p < .001$) were

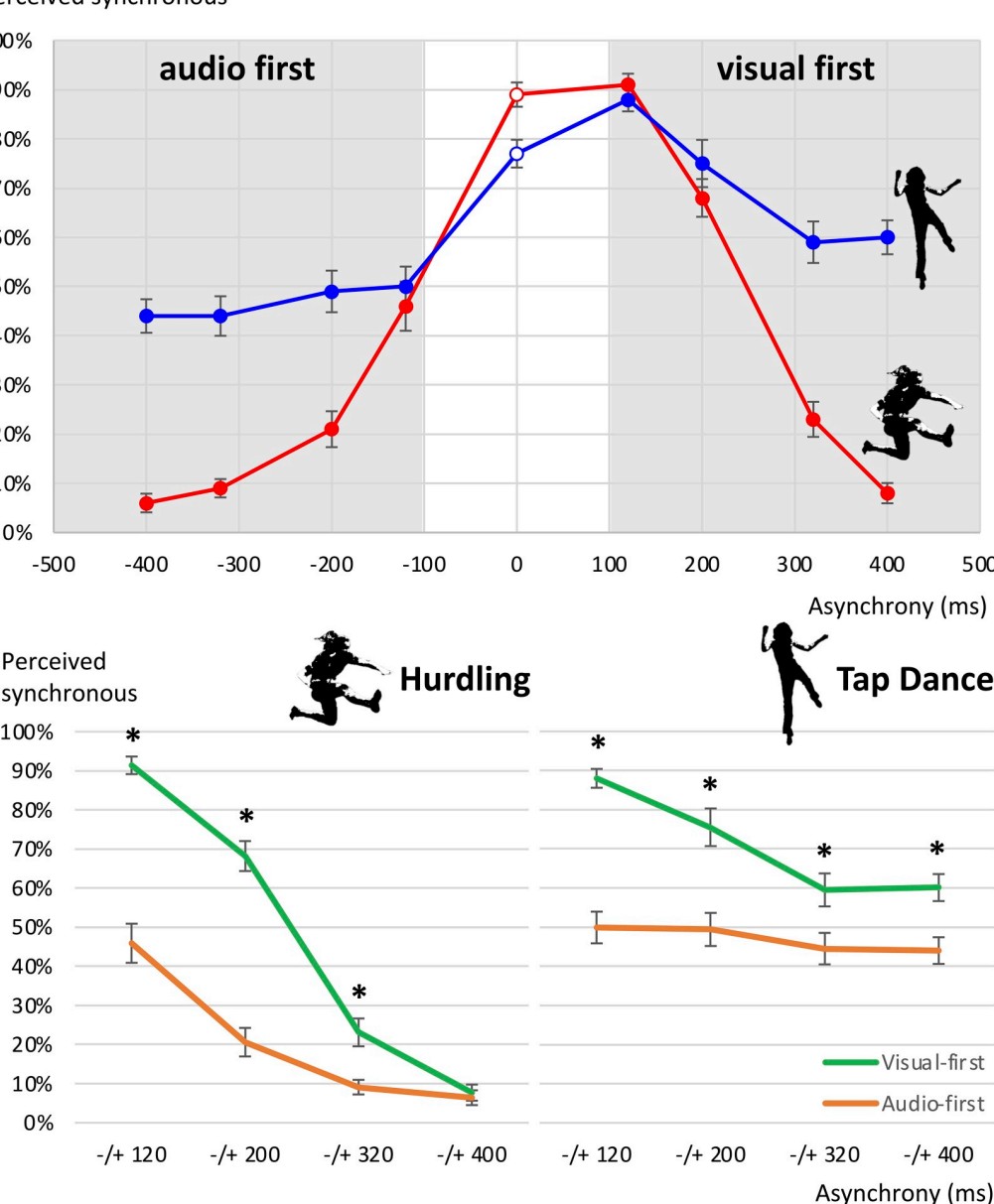

Perceived synchronous

**Fig 4. Mean percentages of trials perceived as synchronous, Study 1.** Asynchronies (in ms) are displayed on the x-axis, with negative values indicating that the auditory channel preceded the visual channel (audio-first) and positive values indicating that the visual channel preceded the auditory channel (visual-first). Error bars show standard error of the mean. The upper panel shows all scores, fanned out for the level combinations of the factors asynchrony size, asynchrony type, and action type. The lower panel illustrates the significant action x asynchrony size x asynchrony type interaction.

both significant, whereas the interaction of ACTION TYPE and ASYNCHRONY TYPE was not ($F(1,21)$ = 1.75, $p$ = .20).

## Interim discussion—Study 1

A consistent finding over all studies examining audiovisual asynchrony processing is that perceived synchrony rates of visual-first conditions are higher compared to audio-first conditions

[11]. Study 1 corroborated this finding for both hurdling and tap dancing stimuli, suggesting that asynchrony perception does not fundamentally differ for whole-body movements. However, perceived synchrony of visual-first compared to audio-first asynchronies was found for larger asynchrony sizes in the tap dancing condition compared to the hurdling condition, as we expected. That is, in tap dancing, audio-first and visual-first perceived synchrony ratings were not only significantly different from each other in the smaller delay types (120 ms, 200 ms), but also in the larger ones (320 ms, 400 ms), whereas in the hurdling conditions, the same difference was found for the 120 ms, 200 ms and 320 ms conditions, but not for the 400 ms condition. This aligns with our assumption that intentionality of sound production leads to differences in the perception of tap dancing and hurdling [28]. We suggest this finding to reflect a wider temporal integration window for our tap dancing condition compared to our hurdling condition. The same mechanism might be at work whenever the temporal integration window for language or music are compared to those for object-related action-induced sounds. For instance, Eg and Behne [24] found a wider temporal integration window for language and music than for chess playing.

Our findings suggest that whole-body movement synchrony perception does not principally differ from other previously examined types of synchrony perception. At the same time, they also point to differences in synchrony perception depending on the intentionality of the produced sounds, with intentional sounds generally being perceived as more synchronous with their visual actions, or having a higher acceptance range, compared to action-induced sound occurring only incidentally.

These results also suggest diverging effects of audiovisual asynchrony on action perception and action execution. In the case of action execution, visual-first asynchronies, i.e. temporal delays of sound, have a disruptive effect on the execution of speaking [33], singing [34] and playing a musical instrument [35], but not on the execution of hurdling [36]. In the case of action perception, on the other hand, those same phase shifts are accepted as synchronous more often in language and music compared to simple object actions [22, 24]. Thus, while asynchronies seem to disrupt action execution for actions intentionally creating sounds, asynchronies for these actions are usually integrated even for relatively large temporal offsets in action perception. Considering that self-initiated sounds during action execution are usually attenuated when compared to externally generated sounds (e.g. [37–40], most likely due to the fact that they are expected [41], the disruption of action execution through experimentally induced audiovisual asynchronies might reflect a heightened sensitivity for unexpectedly delayed sounds in self-performed vs. only observed action.

In sum, Study 1 suggests that characteristics of audiovisual integration in the perception of speech and music may generalize to other types of intentionally sound-generating actions but not to those which create sounds rather incidentally.

Unexpectedly, asynchrony was generally more accurately judged for hurdling than for tap dancing, as reflected by a significant main effect. While this finding does not relativize the reported evidence for a widened temporal window of integration in tap dancing, as suggested by the 400 ms lag condition, it motivates the assumption that it was also more difficult to detect audiovisual asynchrony in tap dancing than in hurdling. Building on these findings in Study 2, we turned to *event density* and *rhythmicity* as factors potentially modulating audiovisual integration; specifically, we sought to test whether they confound our experimental conditions in Study 1. As outlined in the Methods section, we performed a Matlab-based acoustic feature extraction to objectively quantify event density and rhythmicity based on which we conducted the following two post hoc analyses.

Firstly, tap dancing trials had a higher event density (ranging from 3.19 to 4.18; M = 3.74 Hz, SD = .41) compared to hurdling trials (ranging from 2.19 to 2.99; M = 2.69 Hz, SD = .35;

Mann-Whitney-U-test: $U = 0.00$, exact $p = .029$). Event density, i.e. the number of distinguishable auditory sounds occurring per second, or the frequency of distinct events has an influence on the detection performance of audiovisual asynchrony. Visual speech, for example, is integrated roughly at the syllable rate of 4–5 Hz [2, 42, 43]. Temporal frequencies above 4 Hz seem to be difficult to rate in terms of their (a)synchrony [44]. In light of these findings, we post-hoc investigated the effect of event density on synchronicity judgments. To this end, we included EVENT DENSITY as ordinal variable to replace ACTION TYPE in our original ANOVA. This analysis showed a main effect of EVENT DENSITY ($F(2.9, 61.8) = 71.64$, $p < .001$, Greenhouse-Geisser corrected; $\chi^2(14) = 40.60$, $p < .001$, $\varepsilon = .59$), which could mirror our reported main effect of ACTION TYPE. Bonferroni-corrected post-hoc pairwise comparisons of the event density levels showed that differences in performance levels, however, did not mirror the separation point between actions. Instead, no difference in performance was found between the four hurdling videos and one tap dancing video (all $p \geq .52$), which were all lower in performance than the three tap dancing videos with the highest event densities (all $p < .001$), while the video with the highest event density again significantly differed from all others (all $p < .001$). To see whether EVENT DENSITY fully explains the original effect of ACTION TYPE, we calculated another ANOVA including ACTION TYPE and EVENT DENSITY (as ordinal variable within an action) as well as ASYNCHRONY TYPE and ASYNCHRONY SIZE. Here, we found significant main effects of both EVENT DENSITY ($F(2, 42) = 71.09$, $p < .001$) and ACTION TYPE ($F(1, 21) = 74.26$, $p < .001$) as well as their interaction ($F(2, 42) = 68.69$, $p < .001$). These findings suggested that higher synchronicity ratings of tap dancing (vs. hurdling) could not be explained by higher event density in this action type. Moreover, event density did not have the same effect on judging audiovisual synchrony of intentionally and incidentally generated action sounds. As these were post-hoc analyses, we do not further elucidate the other main and interaction effects here. All in all, a direct experimental manipulation and investigation of event density as variable was motivated by these data patterns.

Secondly, tap dancing trials were less rhythmically structured compared to hurdling sounds. Although the overall amplitude of the soundtrack was balanced (i.e. adjusted) between the tap dancing and the hurdling condition, loudness of steps was less variable within tap dancing as compared to hurdling. As the latter was accentuated by three heavy landing steps after hurdle clearance embedded in a sequence of lighter running steps, hurdling might have also led to a more structured percept than tap dancing sounds. A post-hoc Mann-Whitney-U-test showed that the measure of rhythmicity that we explored, the mean amplitude variation coefficient, was lower in tap dancing (M = .47, SD = .10) than in hurdling (M = .91, SD = .14; $U = 0.00$, exact $p = .029$). This is, the four lower mean amplitude variation coefficients allocated to the tap dancing stimuli and the four higher mean amplitude variation coefficients allocated to the hurdling stimuli. In line with the post-hoc analyses for event density reported above, we investigated the effect of rhythmicity operationalized as the mean amplitude variation coefficient on synchronicity judgments in Study 1. The ANOVA including RHYTHMICITY (8), ASYNCHRONY TYPE (2) and ASYNCHRONY SIZE (4) revealed a significant main effect of RHYTHMICITY ($F(3.5, 73.1) = 43.62$, $p < .001$, Greenhouse-Geisser corrected; $\chi^2(27) = 74.27$, $p < .001$, $\varepsilon = .50$). Bonferroni-corrected post-hoc pairwise comparisons showed that the stimuli with the five highest mean amplitude variation coefficients (all hurdling stimuli and one tap dancing stimulus) did not differ in their synchronicity judgments (all $p \geq .968$) but were significantly lower than the three stimuli with the lowest mean amplitude variation coefficients. Within those three stimuli the second lowest differed significantly from the first and the third (all $p \leq .042$). Here, again, the main effect does not mirror the separation point between actions. To see whether RHYTHMICITY fully explains the original effect of ACTION TYPE, we calculated an ANOVA including ACTION TYPE (2), RHYTHMICITY (4), ASYNCHRONY TYPE (2) and ASYNCHRONY SIZE (4). Just as

we found for event density, this analysis showed a main effect of ACTION TYPE ($F(1,21) = 66.18$, $p < .001$), a main effect of RHYTHMICITY ($F(3,63) = 28.47$, $p < .001$) and the interaction of both ($F(3,63) = 33.79$, $p < .001$). These findings suggested that rhythmicity neither had the same effect on intentionally and incidentally generated action sounds. As these were post-hoc analyses, we do not further elucidate the other main and interaction effects here. Results of Study 1 gave rise to the direct experimental manipulation and investigation of rhythmicity, further motivated by the fact that to our knowledge, there is so far no study examining the impact of rhythmicity on perception of audiovisual synchrony.

To summarize these considerations, we found in Study 1 that tap dancing stimuli received generally higher audiovisual synchrony ratings than hurdling stimuli. Since tap dancing videos differed from hurdling videos also with regard to higher event density and lower rhythmicity, both factors were potential sources of confound. To address the potential impact of these factors on audiovisual integration, we conducted Study 2, in which we employed PLDs of drumming sequences with variable event density and rhythmicity. Employing drumming PLDs enabled a direct control of event density and rhythmicity in an otherwise natural human motion stimulus. Note that using drumming actions, we kept intentionality of sound production constant while varying event density and rhythmicity as independent experimental factors. Since PLD markers were restricted to the upper body of the drummer, and since sounds were produced by handheld drumsticks in Study 2 as in contrast to sounds produced by feet in Study 2, we refrained from directly comparing conditions from Study 1 with Study 2.

## Study 2

We recorded PLDs of drumming actions which matched and re-combined parameters of the event density and rhythmicity of the stimuli used in Study 1. Four conditions were generated by instructing the drummer to generate one sequence matching the original hurdling condition in Study 1 (low event density, high rhythmicity, labelled D-R+ hereafter), another matching the original tap dancing stimuli (high event density, low rhythmicity, D+R-), and two sequences with new level combinations of these factors (low event density, low rhythmicity, D-R-, and high event density, high rhythmicity, D+R+).

To investigate whether high event density and low rhythmicity are relevant factors for the temporal binding of multisensory percepts, we applied the same synchrony rating judgment task to our four different classes of drumming stimuli. Based on results from Study 1, we expected that for all stimuli, synchrony ratings are higher for short asynchronies compared to longer asynchronies (120 ms > 200 ms > 320 ms > 400 ms, Hypothesis 1) and visual-first asynchronies to be perceived as synchronous more often than their respective audio-first asynchronies (Hypothesis 2). Regarding the newly introduced factors of event density and rhythmicity, we tested whether higher synchrony ratings are observed for higher event density (Hypothesis 3) and lower rhythmicity (Hypothesis 4).

## Materials and methods–Study 2

Many details regarding participants, the stimulus material and the procedure were the same as in Study 1. Therefore, we here only report aspects that were different between the two studies.

### Participants

The sample consisted of 31 right-handed participants (2 males, 29 females) with an age range from 19 to 29 years ($M = 24.0$, $SD = 2.7$), and all of them were right-handers, as obtained by personal questioning. We recruited only participants who never had a training in drumming. Participants signed an informed consent explaining the procedure of the experiment and the

anonymity of the collected data. Participants studying psychology received course credit for their participation. The study was approved by the Local Ethics Committee at the University of Münster, Germany, in accordance with the Declaration of Helsinki.

## Stimuli

The stimuli used in this study were PLD of drumming actions with matching sound, performed by a professional drum teacher. As in Study 1, PLD were recorded using the Qualisys Motion Capture System and in-ear microphones. Fifteen markers were placed symmetrical on the left and the right shoulders, elbows, and wrists, and on three points of the drumstick and three points of the drum (Fig 1B; exemplary videos can be found in the Supplementary Material). Further processing steps of the video material matched those for Study 1. Finally prepared videos had an average length of 6 s for each of the four factor level combinations (i.e., *D-R+*, *D+R-*, *D-R-*, *D+R+*), with the length of the videos varying from 4.9 s to 6.8 s (M = 5.9 s). The final stimulus set used here consisted of four different types of drumming videos with different event density and rhythmicity parameters as outlined above (*D-R+*, *D+R-*, *D-R-*, *D+R+*). For the conditions replicating our previous hurdling and tap dancing stimuli in event density and rhythmicity (*D-R+*, *D+R-*), the drummer was familiarized with these stimuli and asked to replicate them on the drums. For the two new conditions (*D-R-* and *D+R+*), he was asked to play the previously played sequences either less (*D-R-*) or more (*D+R+*) accentuated. For each of these four sub-conditions, four separate videos were selected, each of which was presented at nine different levels of asynchrony of the sound respective to the visual channel (± 400 / 320 / 200 / 120 ms, and 0 ms). Again, negative values indicated that the audio track was leading the visual track (audio-first) and positive values indicated that the visual track was leading the audio track (visual-first), resulting in 144 different stimuli. All videos included a 1000 ms visual fade-in and fade-out.

To ensure that the 16 newly recorded drumming videos implemented the four different factor level combinations (*D-R+*, *D+R-*, *D-R-*, *D+R+*), we used the same MIRtoolbox as in Study 1 to extract core acoustic features. Fig 2 shows that drumming videos successfully implemented the two experimental factors of mean event density (Hz) and rhythmicity (mean amplitude variation coefficient), resulting in the following combinations: D-R+ (D 2.192, R 0.694), D+R- (D 3.264, R 0.215), D-R- (D 2.538, R 0.162) and D+R+ (D 3.191, R 0.772). Thus, videos with a high event density (D+) had an event frequency of 3.23 Hz, those with low density (D-) 2.37 Hz on average. Videos with a high rhythmicity (R+) had a coefficient of amplitude variation of 0.733, whereas videos with a low rhythmicity (R-) had a coefficient of amplitude variation of 0.189.

As in Study 2, we assessed the mean motion energy (ME) score for all drumming videos (see Methods section of Study 1). This approach yielded a mean ME of 1052 for drumming videos, which was slightly lower than the ME for hurdling (1189) and tap dancing (1220) in Study 1 (S2 Fig). A Kruskal-Wallis test by ranks showed no significant difference between motion energy in hurdling, tap dancing and drumming ($\chi^2(2) = 4.2$, $p = .12$).

## Procedure

The experiment consisted of four experimental blocks. Within each of these blocks, each of the 144 stimuli (four *D-R+*, four *D+R-*, four *D-R-*, and four *D+R+* videos, each with nine different levels of audiovisual asynchrony) were presented once, resulting in 576 trials in total. A pseudo-randomization guaranteed that no more than three videos of the same type of asynchrony (audio-first vs. visual-first) were presented in a row to prevent adaptation to one or the

other. Additionally, it was controlled that no more than two videos of the exact same level of asynchrony were presented directly after each other. We employed the same task as in Study 1.

## Experimental design

The study was implemented with a four-factorial within-subject design with the two-level factor EVENT DENSITY (*low, high*) and RHYTHMICITY (*low, high)*, the four-level factor ASYNCHRONY SIZE (*120 ms, 200 ms, 320 ms, 400 ms*) and the two-level factor ASYNCHRONY TYPE (*audio first, visual first*). The dependent variable was the percentage of the trials perceived as synchronous. Correspondingly, a 2 x 2 x 4 x 2 ANOVA was calculated.

## Results–Study 1

Behavioral results are depicted in Figs 5 and 6. Mauchly's test indicated that the assumption of sphericity was violated for ASYNCHRONY SIZE ($x^2(5)$ = 17.93, $p$ = .003, $\varepsilon$ = .71), EVENT DENSITY x ASYNCHRONY SIZE ($x^2(5)$ = 22.53, $p < .001$, $\varepsilon$ = .65), RHYTHMICITY x ASYNCHRONY SIZE ($x^2(5)$ = 14.63, $p$ = .012, $\varepsilon$ = .78), EVENT DENSITY x RHYTHMICITY x ASYNCHRONY SIZE ($x^2(5)$ = 12.52, $p$ = .028, $\varepsilon$ = .78), and ASYNCHRONY SIZE x ASYNCHRONY TYPE ($x^2(5)$ = 22.01, $p$ = .001, $\varepsilon$ = .68). Therefore, degrees of freedom were corrected using Greenhouse-Geisser estimates of sphericity. As expected, we replicated the main effects of ASYNCHRONY SIZE ($F(2.1,63.9)$ = 186.86, $p < .001$) and ASYNCHRONY TYPE ($F(1,30)$ = 149.59, $p < .001$). Synchrony ratings were highest for the 120 ms asynchronies (M = 69.7%, SD = 14.6%), and decreased with increasing asynchronies (200 ms delays, M = 61.8%, SD = 16.9%; 320 ms asynchronies, M = 48.0%, SD = 19.0%; 400 ms asynchronies, M = 39.8%, SD = 18.5%) with significant differences between all adjacent asynchrony sizes (all $p < .001$, Hypothesis 1). Synchrony ratings were also higher for visual-first asynchronies (M = 64.1%, SD = 14.8%) compared to audio-first asynchronies (M = 45.6%, SD = 19.3%, Hypothesis 2).

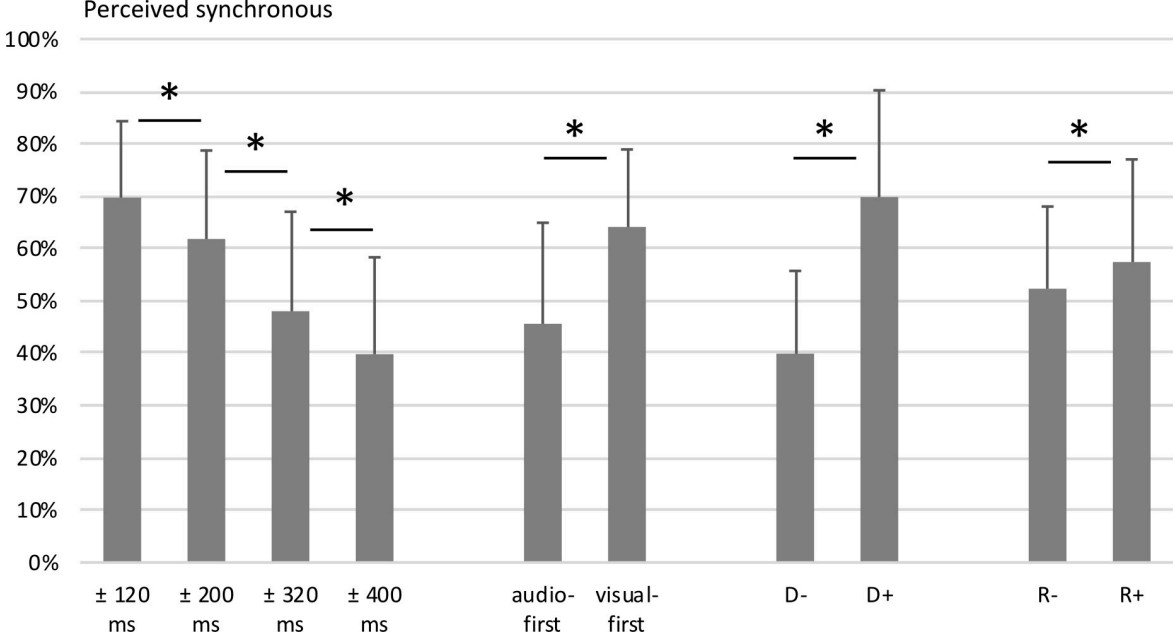

**Fig 5. Main effects of the audiovisual (a)synchrony ratings, Study 2.** Mean percentages of trials perceived as synchronous, aggregated for the factors asynchrony size, asynchrony type, Event density (D- standing for low, D+ for high density) and Rhythmicity (R- and R+ for low and high rhythmicity, respectively). Error bars represent the standard deviation. Significant differences are marked with asterisks.

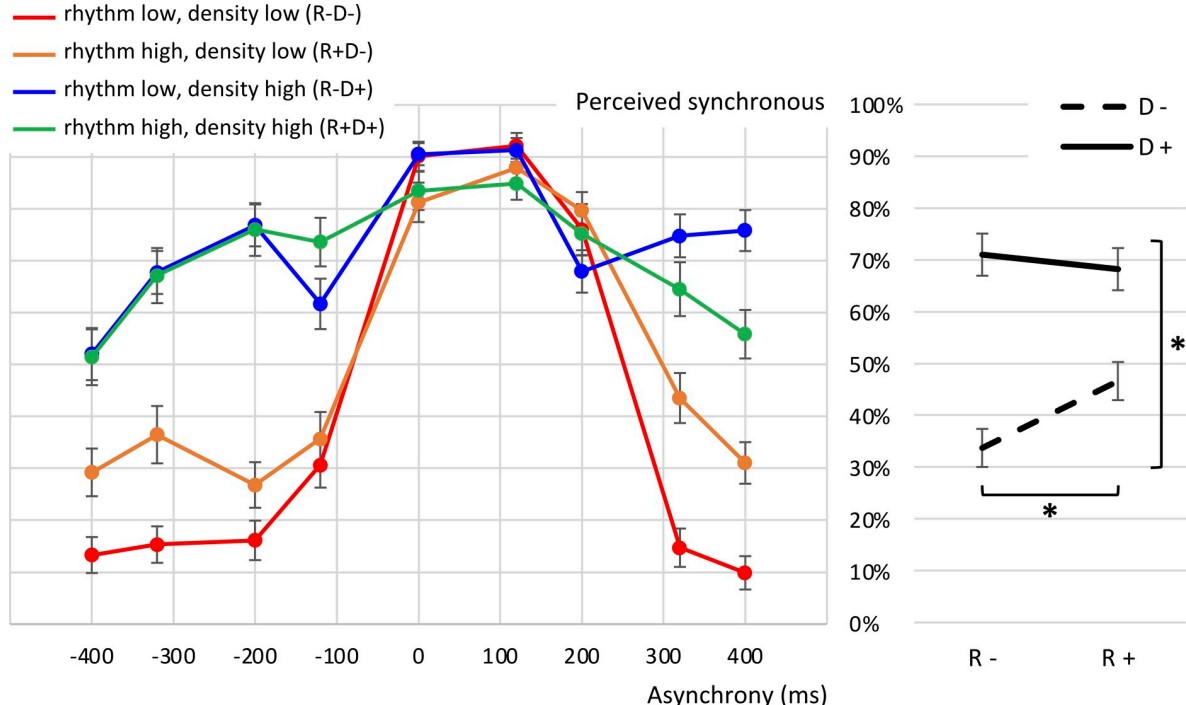

**Fig 6. Mean percentages of trials perceived as synchronous, Study 2.** On the left hand side, all scores are fanned out for the level combinations of the factors asynchrony size, asynchrony type, Event density and Rhythmicity. The right hand side chart illustrates the significant Event Density x Rhythmicity interaction.

We found a main effect for EVENT DENSITY ($F(1,30)$ = 122.30, $p < .001$), with higher event density resulting in higher synchrony ratings (M = 69.8%, SD = 20.4%) compared to lower event density (M = 39.9%, SD = 15.8%, Hypothesis 3). We found a main effect for RHYTHMICITY as well ($F(1,30)$ = 5.48, $p = .026$), but contrary to our hypothesis (Hypothesis 4), synchrony ratings for lower rhythmicity were lower (M = 52.3%, SD = 15.7%) than those for higher rhythmicity (M = 57.4%, SD = 19.6%).

Interaction effects were significant for EVENT DENSITY x RHYTHMICITY ($F(1,30)$ = 22.59, $p < .001$), EVENT DENSITY x ASYNCHRONY SIZE ($F(1.9,58.0)$ = 42.86, $p < .001$), RHYTHMICITY x ASYNCHRONY SIZE ($F(2.3,70.1)$ = 6.26, $p = .002$), EVENT DENSITY x RHYTHMICITY x ASYNCHRONY SIZE ($F(2.3,70.1)$ = 34.63, $p < .001$), EVENT DENSITY x ASYNCHRONY TYPE ($F(1,30)$ = 87.11, $p < .001$), RHYTHMICITY x ASYNCHRONY TYPE ($F(1,30)$ = 4.58, $p = .041$), EVENT DENSITY x RHYTHMICITY x ASYNCHRONY TYPE ($F(1,30)$ = 4.85, $p = .036$), ASYNCHRONY SIZE x ASYNCHRONY TYPE ($F(2.0,61.3)$ = 65.22, $p < .001$), EVENT DENSITY x ASYNCHRONY SIZE x ASYNCHRONY TYPE ($F(3,90)$ = 99.52, $p < .001$), RHYTHMICITY x ASYNCHRONY SIZE x ASYNCHRONY TYPE ($F(3,90)$ = 4.82, $p = .004$), and EVENT DENSITY x RHYTHMICITY x ASYNCHRONY SIZE x ASYNCHRONY TYPE ($F(3,90)$ = 9.76, $p < .001$).

Bonferroni-corrected post-hoc pairwise comparisons inspecting the interaction of EVENT DENSITY and RHYTHMICITY showed significant increases between low and high event densities at both low ($p < .001$) and high ($p < .001$) rhythmicity. Rhythmicity levels increased significantly only for low event density ($p < .001$) but not for high event density ($p = .32$).

## General discussion

Visual and auditory signals often occur concurrently and aid a more reliable perception of events that cause these signals. Audiovisual integration depends on several factors which have

been thoroughly investigated using the example of spoken language or music, but remain largely unexplored regarding their generalizability beyond these domains. In two behavioral studies, we examined the impact of audiovisual stimulus properties that are characteristic for both speech and music, and hence are particularly suited to address the issue of generalizability. In Study 1, we compared audiovisual signals from PLDs which were created intentionally, via tap dancing, and those which were created incidentally, via hurdling. In Study 2, we examined event density and rhythmicity as two properties describing drumming actions and their corresponding sounds.

In line with previous research [11, 22, 24], we found in both Study 1 and 2 that smaller asynchronies tended to be perceived as synchronous more often than larger asynchronies, and that visual-first asynchronies received higher synchrony ratings than their respective audio-first asynchronies. These effects were consistently observed for both, actions that create sounds intentionally as well as incidentally. Interestingly, the average synchrony ratings received for the drumming stimuli (55%) were comparable to those recorded in Study 1 for tap dancing (59%) rather than hurdling (34%), corroborating the interpretation that intentionally action-induced sounds are perceived differently from merely incidentally action-induced sounds. In two previous fMRI study addressing this issue [28, 29], behavioral and brain activity pointed towards stronger auditory expectations (but, importantly, not towards enhanced auditory attention) when observing intentional as compared to incidental sound production. Thus, particularly strong auditory expectations may tend to overrun actual perceptual evidence (e.g. of asynchrony), leading to a pronounced audiovisual integration bias in intentional sound production such as in spoken language. It is well-known that strong prior expectations, while being often helpful for perception, can lead to misperception of degraded sensory signals, causing for instance so-called "slips of the ear" in speech perception and visual illusions [45]. In terms of the predictive coding account, such misperceptions reflect a failure to adjust prior expectations to the current stimulus, either because these prior beliefs are not strong enough or the prediction error is not strong enough [46]. Importantly, part of the generative model is the precision of incoming sensory input and hence potential prediction errors: this expected precision modulates how much the prediction error is weighted in updating predictions. Normally, when we see and hear an actor performing sound-generating movements, sound and sight are synchronous (accepting a slight visual lead because the visual signal always travels slightly faster than the sound). Under normal conditions, when the environment is not particularly noisy, and movement patterns are familiar, the internal models are weighted high and the prediction error relatively low. Temporal regularities, prominent in speech and music, as well as skilled human movement sequences, e.g. tap dancing and hurdling, are particularly powerful cues to enable cross-sensory prediction [47]. Hence, in line with aforementioned fMRI findings on tap dancing and hurdling [28, 29], we suggest that while predictive processes favor the (mis)perception of synchroneity for both hurdling and tap dancing, a more elaborated generative model of to-be-produced sounds may amplify this bias even further. We come back to this assumption below when discussing the effect of rhythmicity.

Study 2 was conducted to figure out whether the effect of intentionality could be partly explained by event density and/or rhythmicity, keeping the intentionality of sound production (in drumming) constant. Here we found that synchrony ratings significantly increased for stimuli with a high event density. That is, the more events occurred per second, the stronger participants were biased towards audiovisual integration—even at very large asynchronies (400 ms). This observation fits well with asynchrony detection collapsing for high event density stimuli, as reported for both speech [2] and audiovisual flash-beep pairings [44]. Petrini and co-workers [18, 48] found that this bias is reduced by practice, as expert drummers outperform novices in audiovisual asynchrony perception for both slow and fast tempi. Importantly, data

of Study 1 suggested that both intentionality and event density had significant effects on increasing the portion of synchronous judgments of actually asynchronous audiovisual stimuli, meaning the effect of intentionality could not be reduced to differences in event density. Since these findings were only derived from a post-hoc comparison treating event density as an ordinal variable, a more reliable test of event density was provided by Study 2, corroborating a main effect of event density on perceived audiovisual synchroneity.

Why does increasing event density disrupt audiovisual asynchrony detection so effectively? A straightforward explanation may be that increasing the event density narrows the width of empty intervals between filled intervals, may they be clicks, tones, or sounds. If we take an average event density of 2.5 Hz, the average onset-to-onset interval between events amounts to 400 ms on average, meaning that a 400 ms asynchrony manipulation shifts the delayed auditory event to coincide with the visual event (or vice versa). Consequently, a 400 ms asynchrony in an audiovisual signal with an event density of 2.5 Hz can only be detected in either of two cases: Either (a), other stimulus features such as amplitude, pitch or spectral frequency are variable enough to indicate a mismatch between the coinciding, phase-shifted visual event (or vice versa). Or (b), the temporal variance between the onset-to-onset intervals is high enough to include longer intervals, bringing a phase-shifted auditory event onto a visual event gap (or vice versa). In the current Study 2, (b) were met by all conditions, and both (a) and (b) in the case of increasing rhythmicity, entailing more variable beat accentuation. For the high event density condition (3.23 Hz), the outcome of higher rhythmicity was negligible. Thus, high event density effectively disturbed asynchrony judgements, independent of the level of rhythmicity. Focusing on this part of the results, one may expect that rhythmicity could only have an effect if event density was not too high, enabling the detection of asynchrony by the fact that the probability of an phase-shifted auditory event to fall into a visual event gap was higher when event density was low enough. And indeed, increasing rhythmicity had a comparably clear effect when event density was low (2.37 Hz).

Contrary to the expected, this higher rhythmicity did not *enhance* asynchrony detection; thus, synchrony ratings were actually lower for more rhythmic trials in Study 2. Note that, while both main effects—density and rhythmicity—were statistically significant, the overall impact on perceived synchronicity was far stronger for density (low: 40% vs. high: 70%) than for rhythmicity (low: 52% vs. high: 57%). Still, the significant interaction of these factors revealed that at the level of low event density, rhythmicity noticeably increased perceived synchronicity from 34% (low rhythmicity) to 47% (high rhythmicity). Two conclusions can be drawn from this finding: first, rhythmicity had a significant effect, but only under the condition that event density did not exceed a certain level. Second, rhythmicity could be ruled out as a confounding factor in Study 1, as tap dancing was less rhythmic than hurdling but lead to higher synchroneity ratings. In other words, the participants' bias to misjudge asynchronous audiovisual PLDs as being synchronous was not explained by the lower rhythmic structure of tap dancing, as compared to hurdling.

To our knowledge, there is no study to date which explicitly examined the influences of temporal structure/rhythmicity on audiovisual asynchrony detection. Thus, it is hard to pinpoint why our more rhythmic stimuli, featuring more distinct events with more discernable moments of impact, led to higher synchrony ratings than the less rhythmic stimuli. At first sight, the opposite would have been plausible, given that, for instance, rhythmicity enhances the detection of auditory stimuli in noisy environments [49]. Since less accurate asynchrony detection has been reported for more complex stimuli in previous studies [4], it is possible that both rhythmicity and density increased overall stimulus' complexity, explaining increased synchrony ratings for both highly dense and highly rhythmic stimuli. However, a worthwhile alternative hypothesis regarding the impact of rhythmicity on audiovisual integration may be

that increasing rhythmicity promotes general *predictability* of the stimulus based on chunking and patterns of accented and unaccented events [50]. Regularity in the stimulus stream and especially a rhythmical event structure is among the most effective sources for temporal predictions [51, 52]. Metrical congruency between the visual and the auditory stream are known to make a slight temporal deviant less noticeable, for instance when we observe dancing to music [53]. Factors promoting predictability therefore may increase our proneness to neglect smaller audiovisual asynchronies [54]. We hence propose that both, intentionally produced sounds and more rhythmically structured sounds, increased (undue) confidence in synchrony. Both effects may be related to increased reliance on top-down predictive models, entailing that asynchronous trials go more often undetected.

## Limitations

Tap dancing, hurdling, and drumming are quite different types of action that may introduce further sources of variance than those we were focusing on in the current studies.

Firstly, the amount of motion may differ between these types of action. The experiments reported here focused on how natural motion sounds are processed. As reported, we assessed this factor in terms of motion energy, and our statistical analysis suggested no significant differences of motion energy in the three tested types of action. Still, the impact of further and more fine-grained parameters describing motion on audiovisual integration, including for instance movement velocity, acceleration, smoothness or entropy (hence predictability) [55] as well as dynamic features related to rhythmic structure in movements [56], remains to be further examined using sound-generating whole-body movements.

Secondly, tap dancing and hurdling may differ with regard to the perceived arousal or the emotional responses they may trigger. In a previous study [28] in which the stimulus set included the videos used in Study 1 we asked participants to indicate whether they found either hurdling or tap dancing more difficult, and to rate the quality of the performance in each single trial. These previous ratings did not reveal any significant differences between tap dancing and hurdling videos. While hurdling and tap dancing are comparable sports in many respects, they differ in terms of expressive or aesthetic appeal. Our studies reported here cannot rule out the influence of this factor, which should be the subject of future investigation.

Finally, one may expect tap dancing videos to increase more auditory attention than hurdling videos. To be sure, in the current study, participants were instructed to deliver an explicit judgement on synchrony of the audiovisual stimuli, obviously entailing attention to both modalities. However, one may speculate that auditory attention would still be higher when we observe tap dancing simply because sounds are produced intentionally in this condition. Two previous fMRI studies including the videos used in Study 1 did not support such an attentional bias for tap dancing [28, 29]. Attention has been found to reverse the typical BOLD attenuation effects observed for predicted stimuli, leading to rather enhanced responses in primary sensory cortices [57]. Contrary to an attentional bias hypothesis, primary auditory cortex was actually significantly and replicable attenuated in tap dancing compared to hurdling in both fMRI studies [28, 29]. These findings are difficult to reconcile with an attentional interpretation of the stronger synchrony bias in tap dancing. Rather, and in line with the fMRI effects, we assume that predictability of the auditory signal plays a crucial role, making intentionally produced sounds more prone to be integrated with their respective visual motion patterns than incidentally produced sounds.

Although it would have been possible to investigate audiovisual integration in the perception of intentionally and incidentally produced sounds using artificially generated stimuli, for the current series of experiments, the focus was on investigating natural movement sounds in

an ecologically valid context. This also had the particular advantage of not biasing subjects' attention in any direction, since we were generating a quite natural perceptual situation. Another approach would have been to combine identical natural actions with intentional and incidental sounds. Here, however, we expected a confound in the sense that subjects would have expected the intentionally produced sound and not the incidental one. Thus, surprise or even irritation effects would have occurred in the incidental condition and would probably have strongly biased the comparison.

## Conclusions

While almost all our physical actions produce sounds, the existing research on audiovisual perception is largely restricted to language and music, and only a handful of studies consider sounds created by object manipulations. However, since speech and music apparently stand out as intentionally produced sounds, it is unclear whether they can be considered as being representatives of action-induced sounds and their audiovisual integration in general. Our present studies contribute to the still very limited number of studies that examine audiovisual integration of natural non-speech stimuli (e.g. [19, 21–24, 48]). Study 1 showed that typical effects reported for audiovisual speech integration extend to the perception of audiovisual asynchrony in whole-body actions, with shorter asynchronies leading to higher synchrony ratings, and an asymmetric temporal integration window favoring integration at visual-first asynchronies. As expected, these effects were even stronger for intentionally as compared to incidentally generated action sounds. Study 2 suggested that high event density effectively disturbs the discrimination of audiovisual asynchronies. As auditory event density of speech excels those achieved by most other types of action-induced sounds, it remains to be investigated whether the considerable bias for integrating asynchronous audiovisual speech stimuli is (at least partly) due to its exceptionally high event density. At low event densities, also stronger rhythmicity increased the overall audiovisual integration bias. We suggest that rhythmicity and intentionality of sound production promote (undue) trust in synchroneity because both foster reliance on a predictive mode of processing. It remains to be tested whether event density and/or rhythmicity have the same effect in incidentally generated action sounds. To clarify this question, and to further our understanding of common principles of audiovisual integration beyond speech and music [17], more research is needed addressing audiovisual integration in incidentally generated action sounds and more real-life audiovisual stimuli, considering the full range of sound features contributing to the variance of audiovisual integration biases.

## Supporting information

**S1 Fig. Example of drumming sequence with high and low rhythmicity (Study 2).** Rhythmicity was operationalized as variation of the amplitude envelope, shown here for two exemplary drumming sequences. While the event density of both recordings is virtually identical (3.42 and 3.39, respectively), the auditory events in the left recording are highly similar in loudness, resulting in low rhythmicity overall (v = 0.25). In contrast, the auditory events within the right recording vary more strongly in loudness, with almost equidistant duplets of loud (i.e. accentuated) events intersected with less accentuated events. This resulted in high rhythmicity overall (v = 0.78).
(TIFF)

**S2 Fig. Motion energy, Study 1 and 2.** The amount of motion quantified by the amount of moving pixels per video for all PLD videos employed to generate different audio-visual

asynchronous stimuli in Study 1 and Study 2. Each black marker depict the motion energy for one video (see Methods of Study 1 for details).
(TIFF)

**S1 Video. Sample video Study 1.** Hurdling, auditory first, 120 ms asynchrony.
(MP4)

**S2 Video. Sample video Study 1.** Hurdling, auditory first, 400 ms asynchrony.
(MP4)

**S3 Video. Sample video Study 1.** Hurdling, visual first, 120 ms asynchrony.
(MP4)

**S4 Video. Sample video Study 1.** Hurdling, visual first, 400 ms asynchrony.
(MP4)

**S5 Video. Sample video Study 1.** Tap dancing, auditory first, 120 ms asynchrony.
(MP4)

**S6 Video. Sample video Study 1.** Tap dancing, auditory first, 400 ms asynchrony.
(MP4)

**S7 Video. Sample video Study 1.** Tap dancing, visual first, 120 ms asynchrony.
(MP4)

**S8 Video. Sample video Study 1.** Tap dancing, visual first, 400 ms asynchrony.
(MP4)

**S9 Video. Sample video Study 2.** Drumming, high event density, high rhythmicity.
(MP4)

**S10 Video. Sample video Study 2.** Drumming, high event density, low rhythmicity.
(MP4)

**S11 Video. Sample video Study 2.** Drumming, low event density, high rhythmicity.
(MP4)

**S12 Video. Sample video Study 2.** Drumming, low event density, low rhythmicity.
(MP4)

## Acknowledgments

We would like to thank Monika Mertens and Marie Kleinbielen for their help during data collection, Theresa Eckes, Alina Eisele, Marie Kleinbielen, and Katharina Thiel for their assistance during filming and with creating the stimulus material, and Niklas Petersen for calculating the motion energy scores. Finally, we would like to thank Nadiya El-Sourani, Amelie Huebner, Klara Hagelweide, Laura Quante, Marlen Roehe and Lena Schliephake for rewarding discussions.

## Author Contributions

**Conceptualization:** Nina Heins, Daniel S. Kluger, Karen Zentgraf, Markus Raab, Ricarda I. Schubotz.

**Data curation:** Stefan Vinbrüx.

**Formal analysis:** Nina Heins, Jennifer Pomp, Stefan Vinbrüx.

**Funding acquisition:** Markus Raab, Ricarda I. Schubotz.

**Investigation:** Nina Heins, Daniel S. Kluger, Stefan Vinbrüx.

**Methodology:** Stefan Vinbrüx.

**Project administration:** Nina Heins.

**Resources:** Karen Zentgraf, Markus Raab, Ricarda I. Schubotz.

**Supervision:** Daniel S. Kluger, Ima Trempler, Axel Kohler, Markus Raab, Ricarda I. Schubotz.

**Validation:** Ima Trempler.

**Writing – original draft:** Nina Heins, Ricarda I. Schubotz.

**Writing – review & editing:** Jennifer Pomp, Daniel S. Kluger, Stefan Vinbrüx, Ima Trempler, Axel Kohler, Katja Kornysheva, Karen Zentgraf, Markus Raab.

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
