## [Decision Letter · Decision Letter 0]

8 Dec 2020

PONE-D-20-30978

Surmising synchrony of sound and sight: Factors explaining variance of audiovisual integration in hurdling, tap dancing and drumming

PLOS ONE

Dear Dr. Schubotz,

Thank you for submitting your manuscript to PLOS ONE. After careful consideration, we feel that it has merit but does not fully meet PLOS ONE’s publication criteria as it currently stands. Therefore, we invite you to submit a revised version of the manuscript that addresses the points raised during the review process.

In particular, I urge the authors to reconsidered the presentation of the rationale and the discussion section, with interpretation of the results. Please submit your revised manuscript by Jan 22 2021 11:59PM. If you will need more time than this to complete your revisions, please reply to this message or contact the journal office at plosone@plos.org. Please include the following items when submitting your revised manuscript:

We look forward to receiving your revised manuscript.

Kind regards,

Alice Mado Proverbio

Academic Editor

PLOS ONE

Journal Requirements:

Reviewers' comments:

Reviewer's Responses to Questions

**Comments to the Author**

1. Is the manuscript technically sound, and do the data support the conclusions?

Reviewer #1: Partly

Reviewer #2: Partly

2. Has the statistical analysis been performed appropriately and rigorously? 

Reviewer #1: Yes

Reviewer #2: Yes

3. Have the authors made all data underlying the findings in their manuscript fully available?

Reviewer #1: Yes

Reviewer #2: Yes

4. Is the manuscript presented in an intelligible fashion and written in standard English?

Reviewer #1: Yes

Reviewer #2: Yes

5. Review Comments to the Author

Reviewer #1: The authors presented two attractive behavioural studies that aimed to investigate audiovisual integration in whole-body actions. They presented the participants with short videos depicting point-light displays from hurdling vs. tap-dancing actions (study 1) and drumming (study 2) actions. The movements could be in synch or out of synch with the produced sounds, and in the latter case, the authors systematically varied the delay between visual and auditory stimulations and their order (video-first vs. audio-first). The participants were instructed to judge the synchronicity of the stimuli explicitly. The results indicated higher synchrony ratings for shorter (vs. longer) asynchronies intervals, for visual stimulation preceding (vs. following) auditory consequences, and for higher (vs. lower) event density. The manuscript is well-organized, the language is appropriate, and the investigation of audiovisual integration in the context of whole-body actions is certainly interesting. That said, I encourage the Authors to take into consideration a series of points that could strengthen the work if integrated with their current proposal. My concerns are specifically focused on the methodological aspects of the studies.

Major points:

- I appreciated that the authors reported possible confounding factors from study 1 and used them to motivate and create Study 2. However, I invite the authors to evaluate further factors that could have impacted or modulated their results.

(1) Did the authors check for a possible confound effect of familiarity of the stimuli? Did the participants have similar familiarity and/or expertise with tap dancing, drumming, and hurdling? The effects of visuomotor expertise (e.g., music, dance, sports) on multisensory processing of whole-body actions are well-documented in the literature.

(2) Did the authors validate their stimuli using a different group of participants before using the videos in their experiment? Were the stimuli comparable in perceived arousal, activation level, or emotional response (e.g., perceived action effort impacts movement feasibility and appreciation judgments)? Did the author consider possible attention effects (e.g., tap dance movements could be more engaging/rewording than hurdling movements)?

- My second point concern the quantification and definition of motion differences between stimulus categories. The authors seem to focus on the acoustic consequences of the actions when creating their categories of stimuli, leaving out the actual movement features. For instance, they based the definitions of "density" on the produced sounds' frequency (e.g., 2.4 Hz vs. 3.4 Hz). There is no information concerning the sequences of movement that produced those sounds. Did the average amount of motion differ between stimulus category? Did the authors take into consideration the different body parts involved in the movements? Did the stimuli differ for kinematic parameters as movement speed and acceleration? There is a documented preference for complex movements characterized by a faster and more complex (vs. slower and uniform) temporal profile (Orlandi et al., 2020).

The authors indicated (lines 386-387) that "For the two new conditions (D-R-, D+R+), he was asked to play the previously played sequences either less (D-R-) or more (D+R+) accentuated". What was the difference in rhythmicity from the kinematic perspective (e.g., variation in the muscular effort or movement acceleration)?

Carrying on with this argument, drumming and hurdling actions involves different body parts (e.g., whole-body vs. upper body) with sounds produced by feet vs. hands, which could introduce a further possible confounding factor. (This comment is consistent with my previous point on observers' expertise and their arousal/emotional response to actions).

I suggest the author provide a more comprehensive rationale and objective quantification of movement density and rhythmicity. It would be good to have a statistical analysis of the kinematics and acoustic features of the different categories of stimuli (objective quantification). Additionally, the author could validate their stimuli by asking participants to rate perceived density and rhythmicity (subjective evaluation) explicitly.

As an example, a recent paper on Cognition introduced motion smoothness/fluency and entropy measures as indices of action timing complexity/rhythmicity and predictability (Orlandi, A., Cross, E. S., & Orgs, G. (2020). Timing is everything: Dance aesthetics depend on the complexity of movement kinematics. Cognition, 205, 104446). Please consider including a reference to the aforementioned work that appears quite crucial in this context.

- My third point concerns the number of trials used in both studies and corresponding analysis. First, the authors report 144 trials for each study, indicating 8 trials per category in study 1 and 4 trials per category in study 2. Secondly, the sample size considered is quite small, especially in study 1. Hence, on the one hand, I suggest the authors provide a rationale for choosing the two sample sizes (e.g., power analysis). On the other hand, ANOVA may not result in the most appropriate statistical method for data analysis. Did the authors check for ANOVA assumption? Maybe, non-parametrical methods will be more sensitive considering the small number of stimuli per condition and sample size. Alternatively, mixed-effects models (e.g., logistic regression) based on single trials (instead of means) may result more effectively.

- I suggest the authors include a limitations paragraph at the end of the discussion section, whether it is not possible to offer a complete explanation or adjustment for the points raised. Furthermore, in light of the above comments, I suggest the authors take into consideration the role of attentional processes, prior expectation, and anticipation (e.g., predictive coding framework) when discussing their results.

Minor points:

- In Figures 3 and 5, please consider reporting statistical significance as p-values or an "*" (with related significance level, e.g., 0.05 reported in the captions).

- Please consider including a figure reporting the kinematic and auditory features of the stimuli categories.

Reviewer #2: This is an interesting study investigating auditory-visual integration in the context of action sounds. The guiding hypothesis is that actions that intentionally produced sound (i.e., where sound is the target of action) should produce greater auditory-visual integration such that sound-action pairs should be perceived as synchronous over a wider range of intervals. This is an interesting idea, but there are many features that differ between both the movements and the sounds that participants were being asked to judge. First of all, as the authors themselves report, sound density differed between the hurdling and tap-dancing stimuli in Study 1. The results of Study 2 which uses different stimuli, appear to confirm the importance of sound density, rather than intentionality.

In addition to sound density, the qualities of the intentionally and unintentionally produced sounds differ acoustically and the movements involved differ in terms of the number of effectors engaged, and the perceived trajectory of actions. All of these features are known to influence auditory-visual integration. (See Chuenn and Schultz, Atten Percept Psychophys (2016) 78:1512–1528; Su, Y.-H. Peak velocity as a cue in audiovisual synchrony perception of rhythmic stimuli. Cognition 131, 330–344 (2014). And Su, YH. Visual tuning and metrical perception of realistic point-light dance movements. Sci Rep 6, 22774 (2016). https://doi.org/10.1038/srep22774 and Vroomens et al. cited in the manuscript).

Finally, the intentional and unintentional sounds differ in another important sense: for intentional sounds the sound is the target of action, whereas for the unintentional sounds the movement sequence (or clearing the hurdle) is the target. Thus the focus of learning and attention in the one case is to form an auditory-motor temporal prediction, whereas the other does not. The same is true of the comparison case of speech, which is discussed at length in the rationale for the experiment.

Together, there seem to be many features that might contribute to perceived auditory-visual synchrony for these stimuli that are not directly aligned with intentionality. While it may not be possible for the authors to address all of these issues with the current data, they need to be considered more carefully I the presentation of the rationale and the interpretation of the results.

Study 1 Questions:

• The authors interpret their results as consistent with the hypothesis that perceived synchrony will be greater for visual-first stimuli for the tap-dancing compared to the hurdling, and that this is because tap-dancing intentionally produces sound. While it is true that the visual-first perceived synchrony is higher, it is also the case that the auditory-first synchrony is higher as well. So, this seems more like an overall effect of task, and doesn’t really seem to fit with the initial hypothesis.

• In the same vein, performance at 0-asynchrony was better for hurdling than tap-dancing showing that the hurdling stimuli are more accurately judged than the tap-dancing stimuli. As with the above, the authors consider this as evidence for the “widened temporal window of integration,” I’m not sure how this can be distinguished from a task difficult effect.

• The authors raise the issue of event density, and examine it in Study 2, but the hurdling stimuli include other possible cues to synchrony such as enhanced visual movement trajectories. Movement trajectory is known to influence perceived timing and auditory-visual integration (See the work of Su, cited above and Vroomens, cited in the manuscript). This issue is addressed in the drumming study, where trajectories across the conditions are more equivalent, and there are no differences at the 0-delay condition.

• The authors hypothesized that event density might affect performance. Was an analysis done to look at the results of Study 1 controlling for event density? If event density does not affect the pattern of results this would be better evidence supporting their hypothesis.

Study 2 Questions:

• The results for the high-density drumming stimuli, which are at a similar rate to the tap-dancing show a very similar pattern of performance. Putting the results of Study 1 and 2 together suggests that the main factor differentiating the audiovisual simultaneity judgements is event density, rather than intentionality. The authors themselves say that the results are consistent with previous findings that simultaneity judgements “collapse” at high event densities.

• The interaction between density and rhythmicity is not described in the Results, only in the Discussion. The authors focus on the main effect of rhythmicity, but this is really driven by the interaction with density.

• The authors try to interpret this finding as indicating that rhythmicity does not affect synchrony judgements for hurdling, but this may not really be true. Running steps are not strictly rhythmic in the way music rhythms are, so it may be hard to compare. I agree that it is not immediately obvious why the more metrically simple stimuli are perceived as more simultaneous, but there may be some sort of “attractor” effect of the beat point. It would be worthwhile to review the literature on beat.

The Discussion is relatively underdeveloped in consists largely in a rehash of the findings. Better integration of the findings with the literature is needed. Also the authors have two previous brain imaging papers using the tap-dancing and hurdling stimuli. It seems like integration of the goals and findings of the current studies with that previous work would make this paper much more substantial.

Minor points:

This sentence is unclear (lines 89-93): “Along these lines, Eg & Behne (24) employed long running and eventful stimuli in their study and concluded that these more natural stimuli can and should be used more in audiovisual asynchrony studies. On the other hand, aberrant audiovisual integration in psychiatric diseases (26) and neurological impairments (27) may well apply beyond speech and music, and thus affect the perception and control of own action.” In the first sentence a more concrete description of the stimuli would be helpful. In the second, it is hard to understand what is meant.

6. PLOS authors have the option to publish the peer review history of their article (what does this mean?). If published, this will include your full peer review and any attached files.

Reviewer #1: No

Reviewer #2: No

---

## [Author Response · Author response to Decision Letter 0]

10 Feb 2021

Dear Professor Proverbio

We are very grateful to you and the reviewers for the constructive feedback regarding our manuscript. We have addressed the reviewers’ valuable comments and suggestions accordingly and wish to submit a revised version of the manuscript for further consideration. Changes to the manuscript have been highlighted and a point-by-point response to the reviewers’ comments can be found below. 

Thank you for your time and consideration. We look forward to hearing from you. 

Yours sincerely,

On behalf of the co-authors

Ricarda Schubotz

Reviewer #1: 

The authors presented two attractive behavioural studies that aimed to investigate audiovisual integration in whole-body actions. They presented the participants with short videos depicting point-light displays from hurdling vs. tap-dancing actions (study 1) and drumming (study 2) actions. The movements could be in synch or out of synch with the produced sounds, and in the latter case, the authors systematically varied the delay between visual and auditory stimulations and their order (video-first vs. audio-first). The participants were instructed to judge the synchronicity of the stimuli explicitly. The results indicated higher synchrony ratings for shorter (vs. longer) asynchronies intervals, for visual stimulation preceding (vs. following) auditory consequences, and for higher (vs. lower) event density. The manuscript is well-organized, the language is appropriate, and the investigation of audiovisual integration in the context of whole-body actions is certainly interesting. That said, I encourage the Authors to take into consideration a series of points that could strengthen the work if integrated with their current proposal. My concerns are specifically focused on the methodological aspects of the studies.

Major points:

- I appreciated that the authors reported possible confounding factors from study 1 and used them to motivate and create Study 2. However, I invite the authors to evaluate further factors that could have impacted or modulated their results.

We appreciate the valuable comments and suggestions made by the Reviewer.

(1) Did the authors check for a possible confound effect of familiarity of the stimuli? Did the participants have similar familiarity and/or expertise with tap dancing, drumming, and hurdling? The effects of visuomotor expertise (e.g., music, dance, sports) on multisensory processing of whole-body actions are well-documented in the literature.

We agree with the Reviewer that familiarity and motor expertise would have been a confounding factor. We therefore excluded participants who received a training in tap dancing, drumming or hurdling. Since tap dancing and hurdling are fairly uncommon types of sports in Germany, and drumming classes are also not very common, we had no problems recruiting naïve participants. We incorporate this exclusion criterion in the Methods sections.

(2) Did the authors validate their stimuli using a different group of participants before using the videos in their experiment? Were the stimuli comparable in perceived arousal, activation level, or emotional response (e.g., perceived action effort impacts movement feasibility and appreciation judgments)? Did the author consider possible attention effects (e.g., tap dance movements could be more engaging/rewording than hurdling movements)?

We did not validate the stimulus material in the strict sense according to perceived arousal or activation level. We had some experiences with the stimulus material from the recording sessions (in a different set of participants) and from findings in two previous fMRI studies including a test-retest quality of performance rating (Heins et al., 2020a; 2020b). Based on these experiences and findings we were (and are) confident that there were no differences between hurdling and tap dancing regarding either perceived action effort, attention, or feelings of reward, for the following reasons. In these previous studies, we presented participants with videos showing their own training performance in hurdling and tap dancing and asked them to rate the quality of their own performance on a Likert scale. We suggest that if there were effects of emotional responses or arousal that yielded differences between tap dancing or hurdling, this group of participants would have been particularly prone to show them, possibly even more than our naïve participants in the present studies. However, participants rated their own performance in tap dancing and in hurdling equally positive, suggesting no differences regarding estimates of efforts, rewarding feelings or appreciation of performance. Moreover, we asked this former group whether they found hurdling or tap dancing more difficult during training, and also these ratings yielded no significant differences between hurdling and tap dancing. Based on these previous findings, we did not expect confounding effects of perceived ease of movements in the current study.

Moreover, regarding attentional effects, the fMRI effects in the mentioned precursor studies using the same type of stimulus material did not suggest increased attention when participants perceived and judged tap dancing, as compared to seeing and judging hurdling trials. Attention has been found to reverse the typical BOLD attenuation effects observed in primary sensory cortices for predicted vs. non-predicted stimuli, leading to enhanced rather than attenuated responses (Reznik et al., 2015; Schröger et al., 2015; Wollman and Morillon, 2018). Interestingly, primary auditory cortex was attenuated in tap dancing compared to hurdling, favoring a prediction-caused attenuation over the attention-caused enhancement explanation of our findings, as hypothesized in these previous fMRI studies. 

We include these observations and considerations from our previous studies in the new limitations section.

- My second point concern the quantification and definition of motion differences between stimulus categories. The authors seem to focus on the acoustic consequences of the actions when creating their categories of stimuli, leaving out the actual movement features. For instance, they based the definitions of "density" on the produced sounds' frequency (e.g., 2.4 Hz vs. 3.4 Hz). There is no information concerning the sequences of movement that produced those sounds. Did the average amount of motion differ between stimulus category? Did the authors take into consideration the different body parts involved in the movements? Did the stimuli differ for kinematic parameters as movement speed and acceleration? There is a documented preference for complex movements characterized by a faster and more complex (vs. slower and uniform) temporal profile (Orlandi et al., 2020).

We agree with the Reviewer’s thoughtful point. While we cannot differentiate between more fine-grained movement features such as movement speed and acceleration, and have now added an analysis of the amount of motion in hurdling, tap dancing, and drumming using the motion energy (ME) calculation based on Matlab (please see changes made in the Methods section of Study 1 and the new Supplementary Figure S2). The mean ME were 1052 for drumming, 1220 for tap dancing, 1189 for hurdling. A Kruskal-Wallis test by ranks showed no significant difference between motion energy in hurdling, tap dancing and drumming (�2(2) = 4.2, p = .12). We report this finding in the revised manuscript as well.

Regarding the body parts involved in the movements, this factor was balanced among the two stimulus categories in Study 1, where point light walkers wore the same number of markers, and hence, the entire body was shown in motion in these videos. For Study 2, of course, there were only markers on the upper part of the drummer’s body. In this respect, stimuli used in Study 1 and Study 2 cannot be directly compared. Please note that we did not intend to compare Study 1 and Study 2 directly, but rather to provide a report on two consecutive experiments, each with its own questions and hypotheses. We rephrased some parts of the original manuscript to make this point clearer than we did before.

We also thank the Reviewer for drawing our attention to the recent study of Orlandi and co-workers which we now cite in the revised manuscript. It would be very interesting, indeed, to examine the effects of further and more fine-grained motion properties on perceived synchronicity in sound-generating whole-body movements.

The authors indicated (lines 386-387) that "For the two new conditions (D-R-, D+R+), he was asked to play the previously played sequences either less (D-R-) or more (D+R+) accentuated". What was the difference in rhythmicity from the kinematic perspective (e.g., variation in the muscular effort or movement acceleration)?

As mentioned above, we are not able to assess acceleration in our stimulus material, but we included the drumming videos in the analysis of motion energy, as described above. 

Carrying on with this argument, drumming and hurdling actions involves different body parts (e.g., whole-body vs. upper body) with sounds produced by feet vs. hands, which could introduce a further possible confounding factor. (This comment is consistent with my previous point on observers' expertise and their arousal/emotional response to actions).

We agree with the Reviewer’s point that directly comparing whole-body and upper-body movements would entail a confound. Note that we did not mean to directly compare drumming and hurdling. We compared hurdling with tap dancing (Study 1), and we compared the parameters of event density and rhythmicity within drumming (Study 2). Thus, conclusions were not drawn from a direct comparison of Study 1 and 2. Study 2 was motivated by the fact that hurdling and tap dancing come at different event densities and rhythmicities, and Study 2 was just meant to examine these factors in more detail. We tried to make this point more clearly at the end of the Interim Discussion motivating Study 2.

I suggest the author provide a more comprehensive rationale and objective quantification of movement density and rhythmicity. It would be good to have a statistical analysis of the kinematics and acoustic features of the different categories of stimuli (objective quantification). Additionally, the author could validate their stimuli by asking participants to rate perceived density and rhythmicity (subjective evaluation) explicitly. As an example, a recent paper on Cognition introduced motion smoothness/fluency and entropy measures as indices of action timing complexity/rhythmicity and predictability (Orlandi, A., Cross, E. S., & Orgs, G. (2020). Timing is everything: Dance aesthetics depend on the complexity of movement kinematics. Cognition, 205, 104446). Please consider including a reference to the aforementioned work that appears quite crucial in this context.

While we cannot provide an as sophisticated and deep analysis of rhythmicity, complexity and predictability as presented in the study of Orlandi and co-workers, we now included an objective and more detailed analysis of rhythmicity (on the auditory side) and of the amount of motion (on the visual side). Since the corresponding MATLAB tools also provide an algorithm to calculate event density, we re-analyzed also event density using the same MATLAB tool for reasons of consistency. Please note in this context, that using different tools changed the event density scores a bit but did not change the overall data pattern. We added new paragraphs in the Methods sections (“Acoustic feature extraction: Event Density and Rhythmicity” and “Assessment of motion energy” and new Figures (Fig. 2 and Supplementary Figures S1 and S2) to document these stimulus features.

Since the number of videos based on which we generated the differently synchronized stimuli was very small (8 for Study 1 and 16 for Study 2), event density and rhythmicity entered as ordinal variables in a post-hoc statistical approach which we now present at the end of the Interim Discussion leading to Study 2. This is, we now objectify the motivation of testing these two factors in Study 2. At the same time, these (albeit only post-hoc) tests show that neither event density nor rhythmicity can fully account for the differences between hurdling and tap dancing since we found a main effect for action type in both statistical approaches. In the end, the Reviewer’s thoughtful comments have made our paper more clearly in this regard, hopefully also in the eyes of this Reviewer. We discuss the consequences of these additional insights, and included potential limitations that our studies have in comparison to Orlandi’s work.

- My third point concerns the number of trials used in both studies and corresponding analysis. First, the authors report 144 trials for each study, indicating 8 trials per category in study 1 and 4 trials per category in study 2. Secondly, the sample size considered is quite small, especially in study 1. Hence, on the one hand, I suggest the authors provide a rationale for choosing the two sample sizes (e.g., power analysis). On the other hand, ANOVA may not result in the most appropriate statistical method for data analysis. Did the authors check for ANOVA assumption? Maybe, non-parametrical methods will be more sensitive considering the small number of stimuli per condition and sample size. Alternatively, mixed-effects models (e.g., logistic regression) based on single trials (instead of means) may result more effectively.

Regarding the number of trials we presented, there is a misunderstanding. We apologize for the obviously unclear description and modified the respective sentences in the Methods sections.

We stated in the original manuscript for Study 1 (lines 199+): “Three blocks with the experimental task were presented thereafter. Within each of these blocks, all the 72 stimuli (four hurdling and four tap dancing videos, each with nine different audiovisual asynchronies) were presented twice, resulting in a total of 144 trials.” Thus, a total of 432 trials were presented in Study 1.

For Study 2, we said “The experiment consisted of four experimental blocks. Within each of these blocks, each of the 144 stimuli (four D-R+, four D+R-, four D-R-, and four D+R+ videos, each with nine different levels of audiovisual asynchrony) were presented once.” (lines 396++). Thus, a total of 576 trials were presented in Study 2.

We have rephrased corresponding paragraphs in the revised manuscript.

Regarding the ANOVA assumptions, we re-checked them for both Study 1 and Study 2, and for factors where Mauchly's test indicated that the assumption of sphericity was violated, degrees of freedom were corrected using Greenhouse-Geisser estimates of sphericity. We wish to thank the Reviewer for this comment. Note that the Greenhouse-Geisser correction did not change any of the reported significant effects.

- I suggest the authors include a limitations paragraph at the end of the discussion section, whether it is not possible to offer a complete explanation or adjustment for the points raised. Furthermore, in light of the above comments, I suggest the authors take into consideration the role of attentional processes, prior expectation, and anticipation (e.g., predictive coding framework) when discussing their results.

We included a new limitations paragraph at the end of the General Discussion to clarify aspects that we cannot exclude based on our studies or findings. Moreover, we extended our discussions considering the role of attention and also refer to predictive coding.

Minor points:

- In Figures 3 and 5, please consider reporting statistical significance as p-values or an "*" (with related significance level, e.g., 0.05 reported in the captions).

We modified Fig. 3 and Fig. 5 accordingly; please note that due to an additional Figure (Fig. 2), Fig. 3 and 5 are now labeled Fig. 4 and 6, respectively.

- Please consider including a figure reporting the kinematic and auditory features of the stimuli categories.

We have included two new figures reporting auditory features of the stimuli (Fig. 2 and Supplementary Figure S1) and a new figure reporting motion energy scores for both Studies (Supplementary Figure S2). 

Reviewer #2: 

This is an interesting study investigating auditory-visual integration in the context of action sounds. The guiding hypothesis is that actions that intentionally produced sound (i.e., where sound is the target of action) should produce greater auditory-visual integration such that sound-action pairs should be perceived as synchronous over a wider range of intervals. This is an interesting idea, but there are many features that differ between both the movements and the sounds that participants were being asked to judge. 

We strongly appreciated the principally positive evaluation of this Reviewer and the constructive and helpful comments.

First of all, as the authors themselves report, sound density differed between the hurdling and tap-dancing stimuli in Study 1. The results of Study 2 which uses different stimuli, appear to confirm the importance of sound density, rather than intentionality.

Actually, after having calculated further post-hoc statistics as motivated by the Reviewers, we now even more objectively and clearly find that the effect of intentionality of sound production stands on its own, in addition to the effects of event density and rhythmicity. Please see our more detailed replies to this point below (question number 4 concerning Study 1). 

In addition to sound density, the qualities of the intentionally and unintentionally produced sounds differ acoustically and the movements involved differ in terms of the number of effectors engaged, and the perceived trajectory of actions. All of these features are known to influence auditory-visual integration. (See Chuenn and Schultz, Atten Percept Psychophys (2016) 78:1512–1528; Su, Y.-H. Peak velocity as a cue in audiovisual synchrony perception of rhythmic stimuli. Cognition 131, 330–344 (2014). And Su, YH. Visual tuning and metrical perception of realistic point-light dance movements. Sci Rep 6, 22774 (2016). https://doi.org/10.1038/srep22774 and Vroomens et al. cited in the manuscript).

We agree with the Reviewer that sound quality as well as movement patterns are relevant factors that have to be considered when comparing intentionally and incidentally sound-generating actions.

First regarding sound quality differences, we explained in the original manuscript (please see “Stimuli” in Methods, Study 1) that we took measures to render the sound qualities of tap dancing and hurdling as comparable as possible, using the following approach: “In a first step, stimulus intensities of hurdling and tap dancing recordings were normalized separately. In order to equalize the spectral distributions of both types of recordings, the frequency profiles of hurdling and tap dancing sounds were then captured using the Reaper plugin Ozone 5 (iZotope Inc, Cambridge, United States). Finally, the difference curve (hurdling – tap dancing) was used by the plugin’s match function to adjust the tap dancing spectrum to the hurdling reference.” As a result, the sound quality of hurdling and tap dancing were highly similar. Please also check the sound of the Supplementary Video Material that we provided for the stimuli.

Regarding the second point, the number of effectors engaged in tap dancing exactly matched those engaged in hurdling. Of course, we did not directly compare the drumming condition (Study 2) with tap dancing or hurdling (Study 1), as here, the number of effectors differed, and markers were restricted to the upper body. We include a statement in the revised manuscript to make this point more prominent.

Finally, we agree that the perceived trajectories of the actions were subjectively different. We now included an additional analysis of the videos quantifying the amount of motion by the index of motion energy (please see Methods Section of Study 1 and new Supplementary Figure S2). The mean motion energy score was 1220 for tap dancing, 1189 for hurdling, and 1052 for drumming. In order to check whether the amount of motion had to be considered a confounding factor in our studies, we calculated a Kruskal-Wallis test by ranks. This test did not show significant differences between motion energy in hurdling, tap dancing and drumming (�2(2) = 4.2, p = .12).

Finally, the intentional and unintentional sounds differ in another important sense: for intentional sounds the sound is the target of action, whereas for the unintentional sounds the movement sequence (or clearing the hurdle) is the target. Thus, the focus of learning and attention in the one case is to form an auditory-motor temporal prediction, whereas the other does not. The same is true of the comparison case of speech, which is discussed at length in the rationale for the experiment.

Our interest in potential differences between processing incidental and intentional types of sound-generating actions stems from exactly these considerations. According to both psychological theories on action (e.g. common coding) as well as according to predictive coding accounts, sound is in either case the effect of these action. But is it in the same sense part of the action goal when we compare incidentally vs. intentionally sound-generating actions? In three previous fMRI studies, two of which we have published so far (Heins et al., 2020a and 2020b), we examined the brain activity for incidental and intentional sound production using “normal” stimuli as well as sound-delayed and sound-deprived video recordings. We found that in tap dancing, primary auditory cortex activity is reduced as compared to hurdling, favoring a stronger predictive “cancelling” of the expected sound. Importantly, attention is known to reverse this effect, leading to enhanced activity in primary sensory cortices. Since we found the opposite, we could (in each of these previous studies) clearly rule out that differences between perceptual processing of tap dancing and hurdling are driven by auditory attention. We now more explicitly consider these previous insights in the revised manuscript and in the limitations section.

While in these previous fMRI studies, we asked participants to judge the quality of the performance in hurdling and tap dancing, participants in the present studies were required to explicitly judge whether the soundtrack and the visual video were synchronous. Thus, task instruction should have ensured that attention was necessarily on both the auditory and the visual channel in either type of action.

Together, there seem to be many features that might contribute to perceived auditory-visual synchrony for these stimuli that are not directly aligned with intentionality. While it may not be possible for the authors to address all of these issues with the current data, they need to be considered more carefully in the presentation of the rationale and the interpretation of the results.

We thank the Reviewer for this suggestion and hope that he/she will find the way we address these points in a convincing manner, as detailed in the following. 

Study 1 Questions:

• The authors interpret their results as consistent with the hypothesis that perceived synchrony will be greater for visual-first stimuli for the tap-dancing compared to the hurdling, and that this is because tap-dancing intentionally produces sound. While it is true that the visual-first perceived synchrony is higher, it is also the case that the auditory-first synchrony is higher as well. So, this seems more like an overall effect of task, and doesn’t really seem to fit with the initial hypothesis.

Reconsidering the way we introduced Hypothesis 3 in the Introduction of Study 1, we think that there was a misunderstanding, caused by unclarity of our own phrasing. 

Based on the literature, we expected a visual-first bias (i.e. a general bias of judging visual-first stimuli more often as synchronous as compared to audio-first stimuli) for both action types (Hypotheses 2). Moreover, we expected that this bias vanishes for the longer delays in hurdling but still persists for tap dancing, if it is true that tap dancing is comparable to speech production in having a larger temporal binding window than incidentally produced action sounds (Hypothesis 3). Our data confirmed this assumption, as we found (i) an interaction of ASYNCHRONY SIZE, ASYNCHRONY TYPE, and ACTION TYPE, and more specifically, (ii) a still significant bias towards synchronous judgments for visual-first as compared to audio-first stimuli at the longest asynchrony delay of 400 ms for tap dancing but not for hurdling. Note that Hypothesis 3 did not necessarily imply that the visual-first effect (Hypotheses 2) was generally larger for tap dancing as compared to hurdling. 

For sure, it is true that we found a main effect for action type, and this is also what we reported in the Results section of Study 1. Importantly, this finding motivated Study 2, as also reported in the original manuscript. Please note that this main effect does not level the hypothesized and observed “extended” visual-first bias for the largest asynchronies in tap dancing. Note also that the post-hoc analysis of action type and event density which we provided according to the Reviewers’ suggestion (see below), corroborated a significant main effect of action type.

We now have rephrased parts of the Introduction and checked the entire manuscript to make this point more clearly. We also noted that we unnecessarily repeated the explanation of the three hypotheses of Study 1 in a part of the Methods section (“Design and statistical hypotheses”) that was also partly redundant with the Results section. We therefore modified this paragraph to avoid confusion. The same was true and applied for the corresponding paragraph in Study 2. We hope that in doing so, we could improve the overall clarity of the hypotheses.

• In the same vein, performance at 0-asynchrony was better for hurdling than tap-dancing showing that the hurdling stimuli are more accurately judged than the tap-dancing stimuli. As with the above, the authors consider this as evidence for the “widened temporal window of integration,” I’m not sure how this can be distinguished from a task difficult effect.

This is an absolutely valid suggestion and indeed, finding a main effect for task reflecting that participants were less accurate in judging synchronicity in tap dancing versus hurdling motivated our Study 2 where we tested potential confounds by different levels of event density and rhythmicity. We make this point now more clearly in the Interim Discussion of Study 1. Please note however, that based on a now more objective quantification of rhythmicity and additional post hoc tests on both factors, data even more clearly show that, while event density and rhythmicity explain a part of the observed synchronicity bias in tap dancing, they still do not fully explain the effect of intentionality. For instance, event density had different effects on the synchronicity judgment in tap dancing and in hurdling, as shown by an interaction of these two factors in the post hoc analysis. 

Based on the original and the additional statistical effects, we suggest - as we now also summarize in the abstract and consider in the revised Discussion more clearly – that overconfidence in the naturally expected, that is, synchrony of sound and sight, was stronger for intentional (vs. incidental) sound production and for movements with high (vs. low) rhythmicity, presumably because both encourage predictive processes. In contrast, high event density appears to increase synchronicity judgments simply because it makes the detection of audiovisual asynchrony more difficult. We finally also re-arranged some paragraphs of the discussion to make it more readable after changes and additions. 

Just to be sure, please note that our interpretation of a widened temporal window of integration for tap dancing was not based on the observation that performance at 0-asynchrony was better for hurdling than tap-dancing but on the observation that the visual-first bias was still significant at 400 ms time lag for tap dancing but not for hurdling. Moreover, for the 120 ms visual-first condition, the performance for hurdling was as bad as for tap dancing (cf. Fig. 4 - formerly labeled Fig. 3), so we did not (in the original manuscript) and will not (in the revised manuscript) base any argument on the 0-asynchrony level or the smallest levels of audio-visual-lag. 

• The authors raise the issue of event density, and examine it in Study 2, but the hurdling stimuli include other possible cues to synchrony such as enhanced visual movement trajectories. Movement trajectory is known to influence perceived timing and auditory-visual integration (See the work of Su, cited above and Vroomens, cited in the manuscript). This issue is addressed in the drumming study, where trajectories across the conditions are more equivalent, and there are no differences at the 0-delay condition.

As described above, we now included an additional analysis of the videos quantifying the amount of motion by a “motion energy” index. The mean ME were 1052 for drumming, 1220 for tap dancing, 1189 for hurdling. A Kruskal-Wallis test by ranks showed no significant difference between motion energy in hurdling, tap dancing and drumming (�2(2) = 4.2, p = .12). However, since motion energy certainly does not capture all dynamic features describing movement, we mention this restriction in the new limitations section. 

• The authors hypothesized that event density might affect performance. Was an analysis done to look at the results of Study 1 controlling for event density? If event density does not affect the pattern of results this would be better evidence supporting their hypothesis.

We agree with the Reviewer that a statistical approach to event density was more informative. To post hoc investigate the effect of event density on synchronicity judgments in Study 1, we included event density as ordinal variable to replace action type in our original ANOVA (please see revised Interim Discussion of Study 1, leading to Study 2). This analysis showed a main effect of event density (F(2.9,61.8) = 71.64, p < .001, Greenhouse-Geisser corrected; �2(14) = 40.60, p < .001, � = .59), which could mirror our reported main effect of action type. Bonferroni-corrected post-hoc pairwise comparisons of the event density levels showed that differences in performance levels, however, did not mirror the separation point between actions. Instead, no difference in performance was found between the four hurdling videos and one tap dancing video (all p ≥ .52), which were all lower in performance than the three tap dancing videos with the highest event densities (all p < .001), while the video with the highest event density again significantly differed from all others (all p < .001). To see whether event density fully explains the original effect of action type, we calculated another ANOVA including action type and event density (as ordinal variable within an action) as well as asynchrony type and asynchrony size. Here, we found significant main effects of both event density (F(2,42) = 71.09, p < .001) and action type (F(1,21) = 74.26, p < .001) as well as their interaction (F(2,42) = 68.69, p <.001). These findings suggested that event density did not have the same effect on intentionally and incidentally generated action sounds. All in all, a direct experimental manipulation and investigation of event density as variable was motivated by these data patterns, then implemented in Study 2.

Study 2 Questions:

• The results for the high-density drumming stimuli, which are at a similar rate to the tap-dancing show a very similar pattern of performance. Putting the results of Study 1 and 2 together suggests that the main factor differentiating the audiovisual simultaneity judgements is event density, rather than intentionality. The authors themselves say that the results are consistent with previous findings that simultaneity judgements “collapse” at high event densities.

It is true that event density had a profound effect on audiovisual simultaneity judgments, as we also discuss in our original manuscript. Inspired by comments of both Reviewers, we now additionally conducted a post-hoc statistical analysis on event density as a categorical variable varying in both action types. As a result, we found significant main effects of both event density as well as action type, and a significant interaction of these factors. Hence, event density did not have the same effect on intentionally and incidentally generated action sounds, and the type of action, differing in intentionality, had an effect on its own. It is obvious that this post-hoc analysis cannot provide the same level of evidence regarding the effect of event density as Study 2, since the variance of event density in tap dancing and in hurdling were not really comparable. However, finding main effects for intentionality and event density, as well as an interaction of both, renders it even more valuable to consider both Study 1 and Study 2 in a common paper, and discussing both as factors modulating the perception of audiovisual (a)synchrony.

• The interaction between density and rhythmicity is not described in the Results, only in the Discussion. The authors focus on the main effect of rhythmicity, but this is really driven by the interaction with density.

We now have added the statistical effect of the interaction in the Results section of Study 2. Actually, due to an editing error before submission, we had deleted this paragraph of the Results section, including also all other significant interaction effects. This missing paragraph has been inserted in the Results section now. We cordially thank the Reviewer to point out this shortcoming.

Albeit small, the main effect of rhythmicity was significant. Even more interesting, the effect of rhythmicity was clear when only looking at low event densities. We think that high event densities obscured the specific effects that rhythmicity had on audiovisual (a)synchrony perception. We have modified the General Discussion to make this point more clearly.

• The authors try to interpret this finding as indicating that rhythmicity does not affect synchrony judgements for hurdling, but this may not really be true. Running steps are not strictly rhythmic in the way music rhythms are, so it may be hard to compare. I agree that it is not immediately obvious why the more metrically simple stimuli are perceived as more simultaneous, but there may be some sort of “attractor” effect of the beat point. It would be worthwhile to review the literature on beat.

We are really sorry but we have to admit that we could not figure out to which statement in our manuscript the Reviewer refers to when saying that we interpret this finding as indicating that rhythmicity does not affect synchrony judgements for hurdling. Regarding the fairly strong effect of rhythmicity for (drumming) stimuli with low event density, rising from 34% (low rhythm) to 47% (high rhythm) synchronous judgements, we now extend the discussion to consider the effect of enhanced predictability by increased rhythmicity.

As a side note, it is important to note that hurdling is highly rhythmical (I would describe its classical three-step structure as indeed beat-based, consisting of a dotted crotchet followed by a quaver and a quaver triplet, resulting in a two-two meter), and learning and training of this rhythm is crucial to achieve a good performance. So we would say that rhythm is not less important for hurdling than for tap dancing performance, but the rhythmic structure and periodicity was subjectively and, as shown by the new analysis on rhythmicity provided in the revised manuscript, also objectively much more evident in hurdling videos.

The Discussion is relatively underdeveloped in consists largely in a rehash of the findings. Better integration of the findings with the literature is needed. Also the authors have two previous brain imaging papers using the tap-dancing and hurdling stimuli. It seems like integration of the goals and findings of the current studies with that previous work would make this paper much more substantial.

Following the Reviewer’s suggestion, we tried to improve the integration of our findings with the literature, including reference to our own previous studies on tap dancing and hurdling in fMRI. Note that we also slightly re-arranged some paragraphs of the General Discussion to improve readability and clarity of the interpretations.

Minor points:

This sentence is unclear (lines 89-93): “Along these lines, Eg & Behne (24) employed long running and eventful stimuli in their study and concluded that these more natural stimuli can and should be used more in audiovisual asynchrony studies. On the other hand, aberrant audiovisual integration in psychiatric diseases (26) and neurological impairments (27) may well apply beyond speech and music, and thus affect the perception and control of own action.” In the first sentence a more concrete description of the stimuli would be helpful. In the second, it is hard to understand what is meant.

We have modified this entire paragraph which was indeed somewhat ill-structured.

---

## [Decision Letter · Decision Letter 1]

21 Apr 2021

PONE-D-20-30978R1

Surmising synchrony of sound and sight: Factors explaining variance of audiovisual integration in hurdling, tap dancing and drumming

PLOS ONE

Dear Dr. Schubotz,

Thank you for submitting your manuscript to PLOS ONE. After careful consideration, we feel that it has merit but does not fully meet PLOS ONE’s publication criteria as it currently stands. Therefore, we invite you to submit a revised version of the manuscript that addresses the points raised during the review process.

While reviewer 1 is satisfied with the way you responded to previous queries, you will see that Reviewer 2 has noticed serious methodological problems inherent to the experimental paradigm, that I will ask you to, please. seriously addressed in the revised version of the paper

We look forward to receiving your revised manuscript.

Kind regards,

Alice Mado Proverbio

Academic Editor

PLOS ONE

Additional Editor Comments (if provided):

Reviewers' comments:

Reviewer's Responses to Questions

**Comments to the Author**

1. If the authors have adequately addressed your comments raised in a previous round of review and you feel that this manuscript is now acceptable for publication, you may indicate that here to bypass the “Comments to the Author” section, enter your conflict of interest statement in the “Confidential to Editor” section, and submit your "Accept" recommendation.

Reviewer #1: All comments have been addressed

Reviewer #3: (No Response)

2. Is the manuscript technically sound, and do the data support the conclusions?

Reviewer #1: Yes

Reviewer #3: No

3. Has the statistical analysis been performed appropriately and rigorously? 

Reviewer #1: Yes

Reviewer #3: Yes

4. Have the authors made all data underlying the findings in their manuscript fully available?

Reviewer #1: Yes

Reviewer #3: No

5. Is the manuscript presented in an intelligible fashion and written in standard English?

Reviewer #1: Yes

Reviewer #3: Yes

6. Review Comments to the Author

Reviewer #1: I much appreciated the authors' effort in considering all points raised during the revision process. The new sections and clarifications have increased the reliability and value of the entire manuscript that is now suitable for publication. I wish the authors the best of luck with their future studies.

Reviewer #3: The study addresses an interesting research question. Does perceived audio-visual synchrony depend on the intentionality of producing sounds? Two experiments are reported. In the first experiment tap dancing and hurdling point light videos are presented to participants at different audio-visual asynchronies. The authors observe a smaller window of perceived synchrony for the hurdling than the tap dancing actions. In order to assess the influence of event density and rhythm in their study, the authors then conduct a second experiment manipulating event density and rhythmicity using drumming stimuli and show that both factors significantly influence simultaneity judgements. Overall, the two experiments appear rather unrelated to each other and I am not convinced that the pattern of results is mainly driven by differences in task difficulty (simultaneity judgements are easier to make for hurdling than for drumming or tap dancing). I have two major concerns related to the choice of experimental design and stimuli on the one hand and the lack of quantification of visual features of the actions on the other hand.

1) It does not become clear to me why the authors chose to compare tap dancing and hurdling in the first place, especially if the idea was to compare intentionally vs. accidentally produced action sounds. In order to show that intentionality matters, it would have been necessary to look at identical actions, and combining these with intentional or accidental action sounds. In the present experimental design intentionality is always confounded with the type of action being performed. Any observed effects can therefore be due to the fact that hurdling is different from tap dancing in many ways other than the intentionality of the action sounds. The second experiment addresses two auditory confounding factors (sound rhythm and event density), by introducing a third new action which is drumming. So the study leaves open many other ways in which these three actions differ both conceptually (artistic vs. competitive) and visually (with respect to their movement kinematics for example)

2) The rigorous assessment of auditory features is not matched by an equally rigorous assessment of the visual features of the stimuli, although such data appears to be available given that actions were recorded using motion capture. The authors only report overall motion energy for the videos computed from the video. What about event density and rhythm in the visual domain? Could it be that synchrony during tap dancing and drumming are harder to detect because the movement amplitude of drumming and tap dancing is much smaller than that of hurdling? What about the saliency of cyclical motion or the influence of visual perspective? The difficulty of audiovisual integration does not only depend on saliency of auditory features but also of the visual stimuli, yet an assessment of how tap dancing, drumming and hurdling are different from each other with respect to their visual aspects and their movement kinematics is missing entirely.

7. PLOS authors have the option to publish the peer review history of their article (what does this mean?). If published, this will include your full peer review and any attached files.

Reviewer #1: No

Reviewer #3: No

---

## [Author Response · Author response to Decision Letter 1]

5 May 2021

Dear Professor Proverbio

We are very grateful to you and the reviewer for the constructive feedback regarding our manuscript. Also, we are glad to hear that Reviewer 1 was satisfied with the way we responded to previous queries, and since we did not hear about any further points raised by Reviewer 2, we assume that s/he was also satisfied with the revision. We addressed the new Reviewer 3’s points in a new revision.

Thank you for your time and consideration. We look forward to hearing from you. 

Yours sincerely,

On behalf of the co-authors

Ricarda Schubotz

Reviewer #3

The study addresses an interesting research question. Does perceived audio-visual synchrony depend on the intentionality of producing sounds? Two experiments are reported. In the first experiment tap dancing and hurdling point light videos are presented to participants at different audio-visual asynchronies. The authors observe a smaller window of perceived synchrony for the hurdling than the tap dancing actions. In order to assess the influence of event density and rhythm in their study, the authors then conduct a second experiment manipulating event density and rhythmicity using drumming stimuli and show that both factors significantly influence simultaneity judgements. Overall, the two experiments appear rather unrelated to each other and I am not convinced that the pattern of results is mainly driven by differences in task difficulty (simultaneity judgements are easier to make for hurdling than for drumming or tap dancing). 

I have two major concerns related to the choice of experimental design and stimuli on the one hand and the lack of quantification of visual features of the actions on the other hand.

1) It does not become clear to me why the authors chose to compare tap dancing and hurdling in the first place, especially if the idea was to compare intentionally vs. accidentally produced action sounds. In order to show that intentionality matters, it would have been necessary to look at identical actions, and combining these with intentional or accidental action sounds. In the present experimental design intentionality is always confounded with the type of action being performed. 

Our reply:

We thank the reviewer for this important reference to the apparently still inadequate account of the motivation behind the choice of the types of movement studied. The current experiments built directly on a series of fMRI studies in which we investigated hurdling and tap dancing with respect to the perceptual processing of the sounds they produce. We added a new passage in the manuscript revision in the introduction and in the discussion, in which we hope to motivate the experimental approach more convincingly:

Introduction:

We decided to use two different types of sporting action that allowed us to study the processing of natural movement sounds in an ecologically valid context. This also had the particular advantage that the subjects' attention was not directed in any direction, since we created a completely natural perceptual situation. 

Limitations:

Although it would have been possible to investigate audiovisual integration in the perception of intentionally and incidentally produced sounds using artificially generated stimuli, for the current series of experiments, the focus was on investigating natural movement sounds in an ecologically valid context. This also had the particular advantage of not biasing subjects' attention in any direction, since we were generating a quite natural perceptual situation. Another approach would have been to combine identical natural actions with intentional and incidental sounds. Here, however, we expected a confound in the sense that subjects would have expected the intentionally produced sound and not the incidental one. Thus, surprise or even irritation effects would have occurred in the incidental condition and would probably have strongly biased the comparison.

Reviewer #3

Any observed effects can therefore be due to the fact that hurdling is different from tap dancing in many ways other than the intentionality of the action sounds. The second experiment addresses two auditory confounding factors (sound rhythm and event density), by introducing a third new action which is drumming. So the study leaves open many other ways in which these three actions differ both conceptually (artistic vs. competitive) and visually (with respect to their movement kinematics for example)

Our reply:

This reference is also helpful for us because it gives us the opportunity to better motivate the choice of movements studied and to address possible limitations of our studies. 

With regard to visual differences between the compared sports, we would like to point out here that our focus was on the topic of movement sounds. In our first revision, we added an analysis of visual kinetic energy according to the reviewer's instructions and found no significant differences here. In addition, we would like to point out that we used only point-light videos in both experiments and thus controlled for much of the visually irrelevant information. Please also consider furthermore our reply to point 2) raised by the Reviewer.

With regard to the conceptual differences raised by the reviewer, we agree that drumming as well as tap dancing are expressive and music-related actions whereas hurdling is not. Please note that we did not mean to directly compare drumming (Exp. 2) with hurdling or tap dancing (Exp. 1) as pointed out in the previous version of the manuscript: “Employing drumming PLDs enabled a direct control of event density and rhythmicity in an otherwise natural human motion stimulus. Note that using drumming actions, we kept intentionality of sound production constant while varying event density and rhythmicity as independent experimental factors. Since PLD markers were restricted to the upper body of the drummer, and since sounds were produced by handheld drumsticks in Study 2 as in contrast to sounds produced by feet in Study 2, we refrained from directly comparing conditions from Study 1 with Study 2.”

However, to more strongly acknowledge this point we now add another passage in the Discussion as follows:

While hurdling and tap dancing are comparable sports in many respects, they differ in terms of expressive or aesthetic appeal. Our studies reported here cannot rule out the influence of this factor, which should be the subject of future investigation.

Reviewer #3

2) The rigorous assessment of auditory features is not matched by an equally rigorous assessment of the visual features of the stimuli, although such data appears to be available given that actions were recorded using motion capture. 

The authors only report overall motion energy for the videos computed from the video. What about event density and rhythm in the visual domain? Could it be that synchrony during tap dancing and drumming are harder to detect because the movement amplitude of drumming and tap dancing is much smaller than that of hurdling? What about the saliency of cyclical motion or the influence of visual perspective? The difficulty of audiovisual integration does not only depend on saliency of auditory features but also of the visual stimuli, yet an assessment of how tap dancing, drumming and hurdling are different from each other with respect to their visual aspects and their movement kinematics is missing entirely.

Our Reply:

It is correct that the rigorous assessment of auditory features is not matched by an equally rigorous assessment of the visual features of the stimuli. While research on motion perception, not only in sports, has for many years focused heavily on and been limited to the visual domain, the focus of our research is clearly on how natural motion sounds are processed. We do not mean to deny the relevance of the visual domain, but to point out that the experiments we present relate specifically to motion sounds. In the first revision of our manuscript, in response to reviewer comments, we also included and added a descriptive and statistical analysis of visual stimulation. We point out this specific focus again in the revised discussion.

---

## [Decision Letter · Decision Letter 2]

31 May 2021

Surmising synchrony of sound and sight: Factors explaining variance of audiovisual integration in hurdling, tap dancing and drumming

PONE-D-20-30978R2

Dear Dr. Schubotz,

We’re pleased to inform you that your manuscript has been judged scientifically suitable for publication and will be formally accepted for publication once it meets all outstanding technical requirements.

Kind regards,

Alice Mado Proverbio

Academic Editor

PLOS ONE

Additional Editor Comments (optional):

The reviewers and I found that all previous comments were successfully addressed by your last revision.

We particularly appreciated your further clarifications relative to the stimulus choice.

Reviewers' comments:

Reviewer's Responses to Questions

**Comments to the Author**

1. If the authors have adequately addressed your comments raised in a previous round of review and you feel that this manuscript is now acceptable for publication, you may indicate that here to bypass the “Comments to the Author” section, enter your conflict of interest statement in the “Confidential to Editor” section, and submit your "Accept" recommendation.

Reviewer #3: All comments have been addressed

2. Is the manuscript technically sound, and do the data support the conclusions?

Reviewer #3: Yes

3. Has the statistical analysis been performed appropriately and rigorously? 

Reviewer #3: Yes

4. Have the authors made all data underlying the findings in their manuscript fully available?

Reviewer #3: Yes

5. Is the manuscript presented in an intelligible fashion and written in standard English?

Reviewer #3: Yes

6. Review Comments to the Author

Reviewer #3: I appreciate that the authors further clarified their choice of stimuli and further address these issues in the newly revised manuscript.

7. PLOS authors have the option to publish the peer review history of their article (what does this mean?). If published, this will include your full peer review and any attached files.

Reviewer #3: No

---

## [Editor Report · Acceptance letter]

14 Jul 2021

PONE-D-20-30978R2 

Surmising synchrony of sound and sight: Factors explaining variance of audiovisual integration in hurdling, tap dancing and drumming 

Dear Dr. Schubotz:

I'm pleased to inform you that your manuscript has been deemed suitable for publication in PLOS ONE. Congratulations! Your manuscript is now with our production department. 

Kind regards, 

on behalf of

Dr. Alice Mado Proverbio 

Academic Editor

PLOS ONE